# Quantum quench dynamics of the attractive one-dimensional Bose gas via the coordinate Bethe ansatz

**Jan C. Zill[1], Tod M. Wright[1], Karen V. Kheruntsyan[1],**
**Thomas Gasenzer[2,3] and Matthew J. Davis[4,5]⋆**

**1** School of Mathematics and Physics, The University of Queensland,
Brisbane QLD 4072, Australia
**2** Kirchhoff-Institut für Physik, Universität Heidelberg,
Im Neuenheimer Feld 227, 69120 Heidelberg, Germany
**3** ExtreMe Matter Institute EMMI, GSI Helmholtzzentrum für Schwerionenforschung,
64291 Darmstadt, Germany
**4** ARC Centre of Excellence in Future Low-Energy Electronics Technologies,
School of Mathematics and Physics, The University of Queensland,
Brisbane QLD 4072, Australia
**5** JILA, University of Colorado, 440 UCB, Boulder, Colorado 80309, USA

⋆ mdavis@physics.uq.edu.au

## Abstract

**We use the coordinate Bethe ansatz to study the Lieb–Liniger model of a one-dimensional gas of bosons on a finite-sized ring interacting via an attractive delta-function potential. We calculate zero-temperature correlation functions for seven particles in the vicinity of the crossover to a localized solitonic state and study the dynamics of a system of four particles quenched to attractive interactions from the ideal-gas ground state. We determine the time evolution of correlation functions, as well as their temporal averages, and discuss the role of bound states in shaping the postquench correlations and relaxation dynamics.**

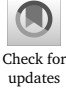
# 1  Introduction

The near-perfect isolation and exquisite control possible for many experimental parameters in ultra-cold atomic gases has enabled the study of nonequilibrium dynamics of closed many-body quantum systems [1]. A number of different trapping geometries have led to the realization of quasi-one-dimensional systems [2–17] that are well described by the paradigmatic exactly solvable Lieb–Liniger model of pointlike interacting bosons [18–20]. As this model is integrable, the various forms of the Bethe ansatz provide powerful methodologies with which to investigate the physics it describes [18, 19, 21–24].

One of the simplest methods of taking a quantum system out of equilibrium is to effect an instantaneous change of a parameter in its Hamiltonian — a so-called quantum quench. Several authors have considered the nonequilibrium dynamics of repulsively interacting systems, where one particularly well-studied scenario is an interaction quench starting from the zero-temperature ideal gas [25–34]. Here we study quantum quenches in which a one-dimensional Bose gas, initially prepared in its noninteracting ground state, is subjected to the abrupt introduction of *attractive* interparticle interactions [35, 36].

The ground-state wave function for the attractive one-dimensional (1D) Bose gas on the infinite line with finite particle number $N$ was constructed by McGuire [37] and consists of a single bound state of all the particles. For systems with finite spatial extent, the coordinate Bethe ansatz provides solutions in terms of quasi-momenta (or rapidities), which for attractive interactions are in general complex-valued. Ground-state solutions on a finite ring were found numerically in Refs. [38, 39].

Since the energy of the ground state is proportional to $-N^3$, where $N$ is the particle number, a proper thermodynamic limit with $N, L \to \infty$ and fixed density $n = N/L$ does not exist [18, 23, 40]. However, the limit $N, L \to \infty$ with $N^3/L = \text{const}$ is well defined, and was recently analysed in Ref. [41]. The zero-density limit $L \to \infty$, $N = \text{const}$ is also well defined and

nontrivial for attractive interactions. In this limit, some correlation functions are accessible with the algebraic Bethe ansatz [42, 43].

An alternative large-system limit is given by $N \to \infty$ in a finite ring of circumference $L$. In particular, in the Bogoliubov limit $c \to 0$, $N \to \infty$, $cN = \text{const}$, where $c$ is the interaction strength, a mean-field Gross–Pitaevskii description of the finite-circumference system predicts the appearance of a localized bright-soliton state beyond some threshold interaction strength [44, 45]. This has been interpreted as evidence for spontaneous breaking of translational symmetry in the infinite-$N$, finite-$L$ limit [44, 46, 47]. However, Bogoliubov theory predicts a diverging quantum depletion in the vicinity of the threshold interaction strength, invalidating the mean-field description in this regime [44].

A many-body analysis for finite $N$ reveals a smooth crossover between a uniform condensate and a state with solitonic correlations, as expected in a finite system [44, 46, 48, 49]. Such an analysis also indicates that the gap at the crossover point vanishes as $N^{-1/3}$ [44]. The Bogoliubov-theory prediction of a vanishing gap at the crossover point in the semiclassical limit $N \to \infty$ is thus regained. The crossover to the correlated state has therefore been interpreted [44] as a kind of effective quantum phase transition in the finite-$L$ system, though it should be stressed that the crossover in a system of finite particle number $N$ cannot be considered a finite-size precursor of a true quantum phase transition, as no proper thermodynamic limit exists.

In a full many-body quantum-mechanical treatment, energy eigenstates on the localized side of the crossover respect the symmetry of the Hamiltonian, but may contain solitonic structure in (pair) correlations. Localized bright solitons can thus be constructed from superpositions of certain exact many-body wave functions [50–52], which are given by the Bethe ansatz [18, 19, 37]. An integral equation for the density of Bethe rapidities of the ground state for particle number $N \to \infty$, valid across the crossover, has recently been derived and signatures of the crossover were observed in this density [47]. Bright-soliton-like structures have also been observed experimentally in elongated quantum-gas samples [53–59].

A particular *nonequilibrium* scenario for the attractive 1D Bose gas was proposed in Refs. [60, 61] and subsequently realized experimentally in Ref. [7]. In the latter work the system was prepared near the ground state at strong repulsive interactions, before the interactions were suddenly switched to strongly attractive using a confinement-induced resonance [20]. In doing so a metastable state was created: the so-called super-Tonks gas [60–64]. This highly excited state of the attractive gas has a "fermionized" character [62] that both stabilizes it against decay via recombination losses and implies a large overlap with the Tonks–Girardeau-like prequench state, leading to efficient state preparation via the interaction quench [63, 64]. This comparatively tractable regime also allows for a Luttinger-liquid description [65], as well as numerical studies with algebraic Bethe-ansatz [65] and tensor-network methods [66]. Local correlations in the super-Tonks regime can be obtained via an identification of the Lieb–Liniger gas with a particular nonrelativistic limit of the sinh-Gordon model [67], as well as by combining the equation of state of the super-Tonks gas with the Hellmann–Feynman theorem [63].

There are fewer results available for more general quench scenarios of the one-dimensional Bose gas involving attractive interparticle interactions. References [68, 69] introduced a Bethe-ansatz method, based on the Yudson contour-integral representation [70], for calculations of nonequilibrium correlation functions in systems of a few repulsively or attractively interacting particles in the infinite-volume limit. Recently, the local second-order correlation function in the relaxed state following a quench from the ideal-gas ground state to attractive interactions was determined in the thermodynamic limit[1] [35, 36] using the quench-action method [71, 72].

---

[1]The quench from the ideal gas to attractive interactions leaves the system with a finite energy per unit length and the thermodynamic limit is therefore well defined in this case [35, 36].

In Refs. [32, 33] we developed a methodology for the calculation of equilibrium and nonequilibrium correlation functions of the repulsively interacting Lieb–Liniger gas based on the semi-analytical evaluation of matrix elements between the eigenstates of the Lieb–Liniger Hamiltonian given by the coordinate Bethe ansatz. Here we extend this approach to the attractively interacting gas, for which the Bethe rapidities that characterize the eigenstates are in general complex-valued, indicating the presence of multiparticle bound states. We apply our method to calculate results for the time evolution of correlation functions following a quench to attractive interactions from the ideal-gas ground state, for a system of four particles. As in our previous studies of quenches to repulsive interactions [32, 33], we find that finite-size effects are significant for quenches to weak final interaction strengths. For strong final interaction strengths our results for the time-averaged local second-order correlation function are consistent with the stationary values in the thermodynamic limit calculated in Refs. [35, 36]. In contrast to that work, however, our approach allows us to also calculate the time-averaged value of the postquench third-order correlation function, which we find to be dramatically enhanced over the ideal-gas value, implying that three-body recombination losses would be significant in experimental realizations of the quench. Our approach also allows us to calculate the dynamical evolution of correlation functions following the quench, and for a quench to strong attractive interactions we observe behaviour similar to that following a quench to repulsive interactions of the same magnitude, superposed with characteristic contributions of bound states at small interparticle separations.

This paper is organised as follows. We provide a brief summary of the Lieb–Liniger model in Sec. 2. We also discuss the complications that arise in numerically solving the Bethe equations due to the appearance of complex Bethe rapidities, and explain how we manage these. In Sec. 3, we calculate ground-state correlation functions for up to seven particles in the vicinity of the mean-field crossover point where solitonic correlations emerge. We also present results for the ground state of four particles subject to strongly attractive interactions. In Sec. 4, we compute representative nonequilibrium correlation functions following quenches of the interaction strength from zero to attractive values for up to four particles. We discuss quenches to the weakly interacting regime in the vicinity of the mean-field crossover, as well as those to the more strongly interacting regime. We also compare the nonequilibrium dynamics to that following an interaction quench to repulsive interactions of the same magnitude. In Sec. 5 we present results for time-averaged correlation functions, before concluding in Sec. 6.

## 2 Methodology

### 2.1 Lieb–Liniger model

The Lieb–Liniger model [18, 19] describes a system of $N$ indistinguishable bosons subject to a delta-function interaction potential in a one-dimensional geometry. The Hamiltonian is

$$\hat{H} = -\sum_{i=1}^{N} \frac{\partial^2}{\partial x_i^2} + 2c \sum_{i<j}^{N} \delta(x_i - x_j), \tag{1}$$

where $c$ is the interaction strength, and we have set $\hbar = 1$ and the particle mass $m = 1/2$. The interactions are attractive for $c < 0$, and repulsive for $c > 0$. The eigenstates of Hamiltonian (1) in the ordered spatial permutation sector $R_p$ ($x_1 \leq x_2 \leq \cdots \leq x_N$) are given by the coordinate

Bethe ansatz in the form [22]

$$\zeta_{\{\lambda_j\}}(\{x_i\}) \equiv \langle\{x_i\}|\{\lambda_j\}\rangle$$

$$= A_{\{\lambda_j\}} \sum_{\sigma} (-1)^{[\sigma]} a(\sigma) \exp\Big[i\sum_{m=1}^{N} x_m \lambda_{\sigma(m)}\Big], \tag{2}$$

where the sum runs over all permutations $\sigma = \{\sigma(1), \sigma(2), \cdots, \sigma(N)\}$ of $\{1, 2, \cdots, N\}$, $(-1)^{[\sigma]}$ denotes the sign of the permutation, and the scattering factors are

$$a(\sigma) = \prod_{k>l} \big(\lambda_{\sigma(k)} - \lambda_{\sigma(l)} - ic\big). \tag{3}$$

The quantities $\lambda_j$ are termed the rapidities, or quasimomenta of the Bethe-ansatz wave function. The normalization constant $A_{\{\lambda_j\}}$ is given by [22]

$$A_{\{\lambda_j\}} = \big[N!\det\{M_{\{\lambda_j\}}\}\prod_{k>l}[(\lambda_k - \lambda_l)^2 + c^2]\big]^{-1/2}, \tag{4}$$

where $M_{\{\lambda_j\}}$ is the $N \times N$ matrix with elements

$$[M_{\{\lambda_j\}}]_{kl} = \delta_{kl}\Big(L + \sum_{m=1}^{N} \frac{2c}{c^2 + (\lambda_k - \lambda_m)^2}\Big) - \frac{2c}{c^2 + (\lambda_k - \lambda_l)^2}. \tag{5}$$

Imposing periodic boundary conditions leads to a set of $N$ equations for the $N$ rapidities, the so-called Bethe equations

$$e^{iL\lambda_j} = \prod_{l \neq j} \frac{(\lambda_j - \lambda_l) + ic}{(\lambda_j - \lambda_l) - ic}, \tag{6}$$

where $L$ is the length of the periodic geometry. The rapidities determine the total momentum $P = \sum_{j=1}^{N} \lambda_j$ and energy $E = \sum_{j=1}^{N} \lambda_j^2$ of the system in each eigenstate. The ground state of the system for attractive interactions is an $N$-body bound state (the finite-system analogue of the McGuire cluster state [37]) and has purely imaginary rapidities [38, 39]. All eigenstates corresponding to bound states have some Bethe rapidities with imaginary components. This is in contrast to the repulsively interacting system ($c > 0$), for which the solutions $\{\lambda_j\}$ to the Bethe equations (6) are purely real. These are usually parameterized by a set of quantum numbers $\{m_j\}$, which for $c \to +\infty$ are proportional to $\{\lambda_j\}$, see e.g. Ref. [22]. For the attractively interacting gas, it is more convenient to enumerate the solutions of the Bethe equations (6) by their corresponding $N$ ideal-gas (i.e., $c = 0$) quantum numbers $\{n_j\}$, where $k_j = 2\pi n_j/L$ are the quantized free single-particle momenta in the finite ring and $n_j$ is an integer [39].[2] In this paper, in which we consider ground-state correlations and quenches from the ideal-gas ground state, we only need to consider eigenstates that are parity invariant, i.e., those for which we can order the $n_j$ such that $n_j = -n_{N+1-j}$ for $j \in [1, N]$. Thus, we can label all eigenstates by $\lfloor N/2 \rfloor$ quantum numbers $\{n_j\}$, where $\lfloor \ldots \rfloor$ is the floor function. By convention we choose these numbers to be the nonnegative values $\{n_j\}$, which we regard as sorted in descending order (for odd $N$, $n_{(N+1)/2} = 0$).

Our results depend explicitly on the number of particles $N$ in our system, though the extent $L$ of our periodic geometry, and consequently the density $n \equiv N/L$ of the gas, is arbitrary.

---

[2]The energy of an eigenstate with $\{n_j\}$ for $c \to 0^-$ connects to the energy of the eigenstate with $\{m_j^{(0)} + n_j\}$ for $c \to 0^+$. Here, $\{m_j^{(0)}\}$ are the quantum numbers of the "Fermi-sea" ground state for $c > 0$. In the remainder of this article, we will label states of the repulsive gas by their reduced quantum numbers $\{n_j\} \equiv \{m_j - m_j^{(0)}\}$.

We follow Refs. [18, 19] in absorbing the density into the dimensionless interaction-strength parameter $\gamma = c/n$. Our finite-sized system is then identified by the specification of both $\gamma$ and $N$. The Fermi momentum $k_F = (2\pi/L)(N-1)/2$, which is the magnitude of the largest rapidity in the ground state in the Tonks–Girardeau limit of infinitely strong repulsive interactions [22], is a convenient unit of inverse length and so we specify lengths in units of $k_F^{-1}$, energies in units of $k_F^2$, and times in units of $k_F^{-2}$.

## 2.2  Correlation functions

The static and dynamic behaviour of the Lieb–Liniger gas can be characterized by the normalized $m^{\text{th}}$-order correlation functions

$$g^{(m)}(x_1,\ldots,x_m,x_1',\ldots,x_m';t) \equiv \frac{\left\langle \hat{\Psi}^\dagger(x_1)\cdots\hat{\Psi}^\dagger(x_m)\hat{\Psi}(x_1')\cdots\hat{\Psi}(x_m')\right\rangle}{\left[\langle\hat{n}(x_1)\rangle\cdots\langle\hat{n}(x_m)\rangle\langle\hat{n}(x_1')\rangle\cdots\langle\hat{n}(x_m')\rangle\right]^{1/2}}, \tag{7}$$

where $\hat{\Psi}^{(\dagger)}(x)$ is the annihilation (creation) operator for the Bose field, $\hat{n}(x) \equiv \hat{\Psi}^\dagger(x)\hat{\Psi}(x)$ is the particle-density operator, and $\langle\cdots\rangle \equiv \text{Tr}\{\hat{\rho}(t)\cdots\}$ denotes an expectation value with respect to a Schrödinger-picture density matrix $\hat{\rho}(t)$. Due to the translational invariance of the system the density is constant [i.e., $\langle\hat{n}(x)\rangle \equiv n$] and the correlation functions are invariant under global coordinate shifts $x \to x+d$. Without loss of generality, we therefore set one of the spatial coordinates to zero and focus on the first-order correlation function $g^{(1)}(x) \equiv g^{(1)}(0,x)$, the second-order correlation function $g^{(2)}(x) \equiv g^{(2)}(0,x,x,0)$, and the local third-order correlation $g^{(3)}(0) \equiv \langle[\hat{\Psi}^\dagger(0)]^3[\hat{\Psi}(0)]^3\rangle/n^3$. We also consider the momentum distribution

$$\tilde{n}(k) = n\int_0^L dx\, e^{-ikx} g^{(1)}(x), \tag{8}$$

which we evaluate at the discrete momenta $k_j$.

For a system in a pure state $|\psi(t)\rangle$, Eq. (7) reads

$$g^{(m)}(x_1,\ldots,x_m,x_1',\ldots,x_m';t) = \frac{1}{n^m}\langle\psi(t)|\hat{\Psi}^\dagger(x_1)\cdots\hat{\Psi}^\dagger(x_m)\hat{\Psi}(x_1')\cdots\hat{\Psi}(x_m')|\psi(t)\rangle,$$

$$= N!\int_0^L \frac{dx_{m+1}\cdots dx_N}{n^m(N-m)!}\psi^*(x_1,\ldots,x_m,x_{m+1},\ldots,x_N,t)\psi(x_1',\ldots,x_m',x_{m+1},\ldots,x_N,t). \tag{9}$$

By expressing the wave function $\psi(\{x_j\},t)$ in terms of Lieb–Liniger eigenstates $\zeta_{\{\lambda_j\}}(\{x_i\})$ [Eq. (2)], we can calculate the integrals in Eq. (9) semi-analytically with the methodology of Ref. [33]. This approach also allows for the evaluation of the overlaps of the initial state with Lieb–Liniger eigenstates necessary for our nonequilibrium calculations in Sec. 4.[3] In Sec. 5, we consider the relaxed state of the system, as described by the diagonal-ensemble [73] density matrix $\hat{\rho}_{\text{DE}} \equiv \sum_{\{\lambda_j\}} \rho_{\{\lambda_j\}}^{\text{DE}}|\{\lambda_j\}\rangle\langle\{\lambda_j\}|$, for which Eq. (7) reads

$$g^{(m)}_{\text{DE}}(x_1,\ldots,x_m,x_1',\ldots,x_m') = \frac{1}{n^m}\text{Tr}\{\hat{\rho}_{\text{DE}}\hat{\Psi}^\dagger(x_1)\cdots\hat{\Psi}^\dagger(x_m)\hat{\Psi}(x_1')\cdots\hat{\Psi}(x_m')\},$$

$$= \frac{1}{n^m}\sum_{\{\lambda_j\}}\rho_{\{\lambda_j\}}^{\text{DE}}\langle\{\lambda_j\}|\hat{\Psi}^\dagger(x_1)\cdots\hat{\Psi}^\dagger(x_m)\hat{\Psi}(x_1')\cdots\hat{\Psi}(x_m')|\{\lambda_j\}\rangle,$$

$$= N!\sum_{\{\lambda_j\}}\rho_{\{\lambda_j\}}^{\text{DE}}\int_0^L\frac{dx_{m+1}\cdots dx_N}{n^m(N-m)!}\zeta^*_{\{\lambda_j\}}(x_1,\ldots,x_m,x_{m+1},\ldots,x_N)$$

$$\times \zeta_{\{\lambda_j\}}(x_1',\ldots,x_m',x_{m+1},\ldots,x_N). \tag{10}$$

---

[3]We note that direct evaluation of the normalization constant $A_{\{\lambda_j\}}$ via Eq. (4) is susceptible to catastrophic cancellations similar to those discussed in Appendix B. In practice, we therefore obtain the constants $A_{\{\lambda_j\}}$ by evaluating the self-overlaps of unnormalized Bethe eigenfunctions using the methodology of Ref. [33].

## 2.3 Numerical considerations

For repulsive interactions the solutions to the Bethe equations (6) are characterized by purely real rapidities $\{\lambda_j\}$, and finding these numerically is relatively straightforward — see, e.g., Ref. [32]. However, for attractive interactions solutions with complex rapidities are possible, and the associated Yang-Yang action [21] of the problem is nonconvex (see, e.g., Ref. [43]), which significantly complicates the root-finding procedure.

To find the rapidities for attractive interactions, we start our root-finding routine close to $\gamma = 0$. Here the rapidities $\{\lambda_j\}$ are close to the free-particle momenta corresponding to $\{n_j\}$, and these can be used as an initial guess for a Newton-method root finder. We then decrease $\gamma$ in small steps, using linear extrapolation of the previous solutions to form initial guesses for the rapidities at each new value of $\gamma$. We have found that this procedure gives good convergence of the rapidities to machine precision.

Eigenstates with complex rapidities arrange themselves in so-called string patterns in the complex plane for large values of $|c|L \equiv N|\gamma|$, with deviations from these strings exponentially small in the system length $L$ [23, 39, 42, 43, 74]. For these states, some of the scattering factors $a(\sigma)$ in Eq. (3) become increasingly smaller with increasing $|\gamma|$, cancelling the extremely large exponential factor to give a finite result. Naïve evaluation of the wave function would therefore lead to numerical inaccuracies due to catastrophic cancellations as soon as the string deviations shrink to the order of machine precision. This problem can be overcome by using the Bethe equations (6) to rewrite the problematic factors in $a(\sigma)$ in terms of exponentials, thereby rendering the expressions more amenable to numerical calculation, as we discuss in Appendix B. For $N = 4$, this enables us to calculate correlation functions for attractive interaction-strength values $\gamma \geq -40$ using standard double-precision floating-point arithmetic, with the exception of a single eigenstate that we treat with high-precision arithmetic, as we discuss in Appendix B.3. For larger values of $|\gamma|$, the bound states become increasingly localized, leading to factors in Eq. (2) that are too large to be represented with double-precision floating-point arithmetic. We could in principle treat systems with $\gamma < -40$ through extensive use of high-precision arithmetic, but find that the regime $\gamma \geq -40$ to which we restrict our analysis reveals many important features of the physics of the attractively interacting system.

## 3 Ground-state correlation functions

The ground-state correlation functions of the one-dimensional Bose gas with attractive interactions have so far been investigated both using mean-field [44, 45, 75] and beyond-mean-field methodologies [44, 46, 48, 49, 76]. The corresponding Bose-Hubbard lattice approximation was considered in Ref. [77]. Systems in the limit $L \to \infty$ were studied in Refs. [40, 51, 78–80], while in Ref. [81] correlation functions for up to $N = 4$ particles under hard-wall boundary conditions were obtained via the coordinate Bethe ansatz. References [42, 43] used the algebraic Bethe ansatz to calculate the dynamic structure factor to first order in the string deviations under periodic boundary conditions. Piroli and Calabrese recently computed the local two- and three-body correlations in the limit where the interaction strength goes to zero as the system size increases at fixed particle density [41].

Here we compute exact correlation functions for a finite system of length $L$ with periodic boundary conditions and compare them with the predictions of mean-field theory, first for $N = 7$ particles in the vicinity of the uniform-density to bright-soliton crossover $-0.7 \leq \gamma \leq 0$, before considering more strongly attractive systems of $N = 4$ particles with $-40 \leq \gamma \leq -2$.

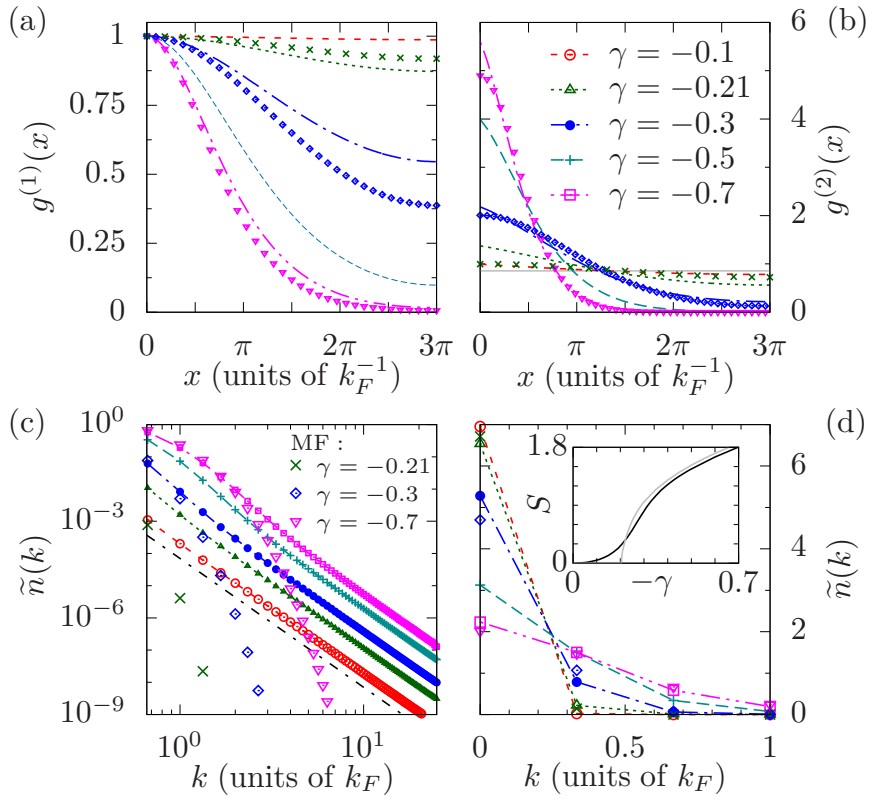

Figure 1: Ground-state correlation functions for $N = 7$ particles and interaction strengths of $\gamma = -0.1, -0.21, -0.3, -0.5$, and $-0.7$. For comparison, we also plot the mean-field correlation functions for $\gamma = -0.21$ (green crosses), $\gamma = -0.3$ (blue diamonds), and $\gamma = -0.7$ (open pink triangles). The mean-field critical interaction strength is $\gamma_{\text{crit}} \simeq 0.201$. (a) First-order correlation $g^{(1)}(x)$ in the spatial domain. (b) Second-order correlation $g^{(2)}(x)$. The horizontal line indicates the result for the noninteracting ($\gamma = 0$) gas. (c) Momentum distribution $\widetilde{n}(k)$. The black dot-dashed line indicates $\propto k^{-4}$ scaling. (d) Momentum distribution $\widetilde{n}(k)$ for small momenta on a linear scale. Inset: Single-particle entanglement entropy $S$ for Bethe-ansatz calculations (black line) and mean-field calculations (grey line).

## 3.1 Correlations near the crossover

In Fig. 1 we plot the first- and second-order correlation functions of the ground state for $N = 7$ particles for a range of $\gamma$. Figure 1(a) shows the first-order correlation $g^{(1)}(x)$ in the spatial domain. For $\gamma = -0.1$ (red dashed line), the proximity to the noninteracting gas results in a nearly constant $g^{(1)}(x)$. For more attractive values of $\gamma$, $g^{(1)}(x)$ begins to decay towards zero at larger separations $x$. For $\gamma = -0.7$ (pink dot-dashed line), $g^{(1)}(x)$ comes close to zero for $x = 3\pi k_F^{-1}$, which corresponds to $x = L/2$ for $N = 7$. [Due to the periodic nature of our geometry, $g^{(1)}(x)$ is symmetric around $x = L/2$, and we therefore only show $g^{(1)}(x)$ up to this point.]

Mean-field theory predicts a crossover from a uniform mean-field wave function to a localized bright-soliton state at an interaction strength $\gamma_{\text{crit}} = -\pi^2/N^2 \simeq -0.201$ [44–47]. In our exact quantum-mechanical treatment of the translationally invariant (and particle-number conserving) system, the density is necessarily constant. However, a signature of the bright-soliton-like state can be found in the first-order correlation function. In the finite-sized system the crossover is broad, but there is clearly a significant change in $g^{(1)}(x)$ between $\gamma = -0.1$ [red dashed line in Fig. 1(a)] and $\gamma = -0.3$ (blue dot-dashed line). In the mean-field description,

the many-body wave function is approximated by a translationally symmetrized Hartree-Fock product of single-particle wave functions [79, 80]. In this approximation correlation functions are comparatively straightforward to compute numerically, see Appendix A for details.

Whereas the mean-field analysis predicts a sharp transition to the localized regime at the threshold interaction strength, the inclusion of quantum fluctuations leads to a smooth crossover between the delocalized and localized regimes in a system of finite $N$ [39, 44]. To characterize the breadth of the crossover in our system, we calculate the single-particle entanglement entropy; i.e., the von Neumann entropy $S = -\text{Tr}\{\rho^{(1)}\log(\rho^{(1)})\}$ of the single-particle density matrix $\rho^{(1)}(x, x') = n g^{(1)}(x, x')$ [82]. In translationally invariant systems $S = -\sum_j [\tilde{n}(k_j)/N] \log[\tilde{n}(k_j)/N]$, where the $\tilde{n}(k_j)$ are the momentum-mode populations.

In the (symmetrized) mean-field description, the ground state for $\gamma > \gamma_{\text{crit}}$ is a pure product state, and hence $S = 0$. For $\gamma < \gamma_{\text{crit}}$, the ground state is a superposition of bright solitons, and $S > 0$ [46]. This can indeed be seen in the inset of Fig. 1(d), where we plot the single-particle entanglement entropy of the exact solution (black line) and of the mean-field solution (grey line) for $N = 7$ particles. The mean-field entropy $S(\gamma)$ exhibits a slope discontinuity at the crossover point, whereas the von Neumann entropy of the exact ground state (black line) varies smoothly.

For $\gamma > \gamma_{\text{crit}}$ the mean-field wave function is uniform, leading to a constant $g^{(1)}(x)$. In Fig. 1(a) we compare our exact results to the mean-field solution just on the localized side of the crossover at $\gamma = -0.21$ (green crosses), and find that the exact many-body solution (green dotted line) is slightly more localized. By contrast, for $\gamma = -0.3$, i.e., further from the crossover point, the mean-field solution (blue diamonds) is more localized than the exact solution (blue dot-dashed line). For $\gamma = -0.7$ the mean-field solution (pink triangles) and the exact $g^{(1)}(x)$ (pink dot-dot-dashed line) are reasonably similar, though the mean-field solution is again somewhat more localized than the exact solution. We note that this behaviour is consistent with that of the entanglement entropy [inset to Fig. 1(d)], which is smaller for the exact solution than for the mean-field approximation for $|\gamma| \gtrsim 0.23$. By contrast, at weaker interaction strengths finite-size rounding of the crossover yields an entropy for the exact system larger than the mean-field value.

In Fig. 1(c), we plot the momentum distribution $\tilde{n}(k)$ corresponding to the first-order correlations shown in Fig. 1(a). [For our system $\tilde{n}(k_j, t) \equiv \tilde{n}(-k_j, t)$ and hence we only plot positive momenta.] We note that for all interaction strengths we consider here, the exact momentum distributions exhibit a power-law decay $\tilde{n}(k) \propto k^{-4}$ at high momenta — the universal large-momentum behaviour for systems with short-range interactions [83–85]. For the case of $\gamma = -0.1$ (red empty circles), interactions are sufficiently weak that no visible deviation from this scaling is visible at the smallest nonzero momenta $k_j$ resolvable in our finite geometry. By contrast, for $\gamma = -0.21$ (green triangles), less trivial behaviour of the momentum distribution can be seen, with the lowest nonzero momentum modes deviating visibly from the $\propto k^{-4}$ scaling. As $|\gamma|$ increases, the deviations from this scaling extend to higher momenta, and a broad hump in the momentum distribution develops. This broadening can be more clearly seen in Fig. 1(d), where we plot the momentum distribution for low momenta $k \leq 1k_F$ on a linear scale. For $\gamma = -0.1$ (red empty circles), the zero-momentum occupancy is close to its ideal-gas value of $\tilde{n}(k = 0) = N$. The zero-momentum mode occupation decreases with increasing $|\gamma|$ and much of this population is redistributed to the first few nonzero momentum modes, resulting in, e.g., a broad distribution $\tilde{n}(k)$ for $\gamma = -0.7$ (pink empty squares).

The ground-state mean-field momentum distributions in Fig. 1(c) do not show the $\propto k^{-4}$ scaling for large $k$ — this feature appears with a first-order Bogoliubov analysis [86]. For an interaction strength $\gamma = -0.21$, i.e., close to the crossover point, the exact $\tilde{n}(k)$ (green dotted line) and the mean-field solution (green crosses) are clearly different away from $k = 0$. For larger attractive values of $\gamma$, however, the two momentum distributions start to agree more

closely. For example, from Figs. 1(c) and 1(d) we observe reasonable agreement between the exact and mean-field results for the lowest three modes at $\gamma = -0.3$ (blue diamonds for mean-field solution, blue dot-dashed line for exact solution). Even closer agreement is observed for $\gamma = -0.7$, where the lowest six modes of the exact solution (pink dot-dot-dashed line) agree well with the mean-field solution (pink triangles), before the $\propto k^{-4}$ tail of the exact momentum distribution takes over.

In Fig. 1(b), we plot the second-order correlation $g^{(2)}(x)$ for the same values of $\gamma$ as before. For $\gamma = -0.1$ (red dashed line), $g^{(2)}(x)$ is close to the ideal-gas value $g^{(2)}_{\gamma=0}(x) = 1 - 1/N$ (horizontal grey line). For $\gamma = -0.21$ (green dotted line), $g^{(2)}(x)$ is increased over the ideal-gas value at distances $x \lesssim 1.3\pi k_F^{-1}$ and correspondingly decreased at larger distances. This behaviour is even more pronounced for $\gamma = -0.3$ (blue dot-dashed line), and the trend continues for larger attractive values of $\gamma$, for which there is significant bunching of particles. Comparing the exact results to the mean-field solutions, we again observe a clear difference at $\gamma = -0.21$, where the exact solution (green dotted line) is more localized than the mean-field solution (green crosses). For $\gamma = -0.3$, the exact solution (blue dot-dashed line) has a slightly increased value at zero separation compared to the mean-field solution (blue diamonds), but at intermediate separations the latter is marginally broader. For $\gamma = -0.7$, the local value $g^{(2)}(0)$ of the exact solution (pink dot-dot-dashed line) is again somewhat larger than the mean-field value (pink triangles). At separations $x \gtrsim \pi/4 \, k_F^{-1}$, the mean-field and exact distributions show good agreement.

## 3.2 Correlations for strongly interacting systems

In Fig. 2, we plot the first- and second-order correlation functions of the ground state for $N = 4$ particles and for a larger range of values of the interaction strength $-40 \leq \gamma \leq -2$. For $N = 4$, the mean-field critical interaction strength is $\gamma_{\text{crit}} \simeq -0.617$, and all ground states we consider here are therefore well in the localized regime. Figure 2(a) indicates the first-order correlation function $g^{(1)}(x)$, which shows that the soliton-like state becomes increasingly tightly localized with increasing $|\gamma|$. This can also be observed in momentum space, Fig. 2(c), where the corresponding momentum distributions $\widetilde{n}(k)$ become broader with increasing $|\gamma|$. We note that the momentum distributions for the most strongly interacting systems considered here are much broader than the "hump" that forms in the ground-state momentum distribution of the repulsive gas in the strongly interacting Tonks limit, which extends to $\simeq 2k_F$ [33, 87, 88]. For comparison, we also plot the mean-field correlation functions for $\gamma = -40$ in Figs. 2(a) and (c) (grey diamonds). The mean-field first-order correlation function is similar to that of the exact solution but slightly more localized, and the momentum distribution is correspondingly somewhat broader than the exact distribution for small values of $k$. Nevertheless, the two momentum distributions agree well over a wide range of momenta up to $k \simeq 30k_F$, where the universal $\propto k^{-4}$ scaling of the exact momentum distribution begins.

Figure 2(b) shows the second-order correlation $g^{(2)}(x)$ for separations up to $x = \pi/4 \, k_F^{-1}$ (which corresponds to $x = L/12$ for $N = 4$). We again observe that the system becomes more tightly bound with increasingly attractive interactions. In order to ensure that the form of the correlation function at moderate separations $x$ is visible in this figure, we have limited the extent of the $y$-axis. The maximum value of the second-order correlation function for $\gamma = -40$ (solid black line), $g^{(2)}(x = 0) = 100$, is therefore not shown. The mean-field correlation function for $\gamma = -40$ (grey diamonds) shows good agreement with the exact solution, though its value at zero separation $g^{(2)}_{\text{MF}}(x = 0) = 80$ (not shown) is reduced compared to that of the exact solution.

Figure 2(d) shows the local second- and third-order correlations for a wide range of interaction strengths. For small values of $|\gamma|$, these correlations are close to their respective ideal-gas

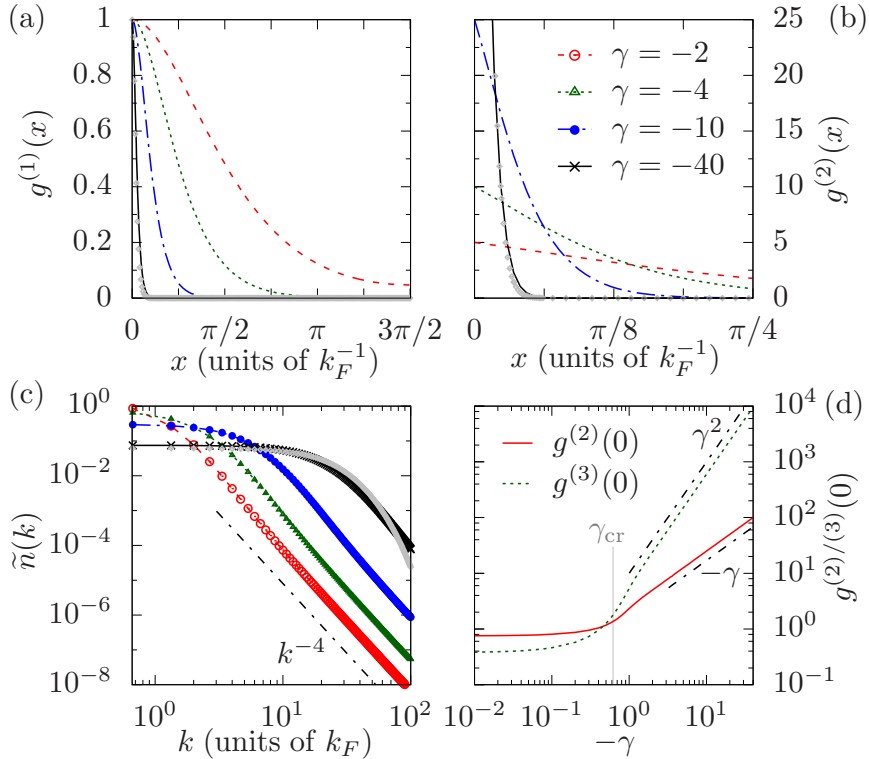

Figure 2: Ground-state correlation functions for $N = 4$ particles and interaction strengths $\gamma = -2, -4, -10$, and $-40$. (a) First-order correlation $g^{(1)}(x)$. (b) Second-order correlation $g^{(2)}(x)$. The local values for $\gamma = -40$, $g^{(2)}(0) = 100$ and $g^{(2)}_{MF}(0) = 80$, exceed the shown range. (c) Momentum distribution $\tilde{n}(k_j)$. Grey diamonds in (a)–(c) correspond to the mean-field solution for $\gamma = -40$. (d) Local second- and third-order correlation $g^{(2)}(0)$ and $g^{(3)}(0)$, respectively, for a range of interaction strengths $\gamma$. Black dot-dashed lines indicate power-law scaling, proportional to $-\gamma$ (lower line) and $\gamma^2$ (upper line).

values, $g^{(2)}(0) = 1 - 1/N = 0.75$ and $g^{(3)}(0) = N(N-1)(N-2)N^{-3} = 0.375$ [89]. In the vicinity of the mean-field crossover point (indicated by the vertical grey line), both $g^{(2)}(0)$ and $g^{(3)}(0)$ begin to increase significantly with increasing $|\gamma|$. For larger values of $|\gamma|$, we observe a linear scaling of the second-order correlation $g^{(2)}(0) \propto -\gamma$ and a quadratic scaling of the third-order correlation $g^{(3)}(0) \propto \gamma^2$, both of which we indicate by black dot-dashed lines in Fig. 2(d). The former scaling can be understood by noting that the McGuire cluster energy scales as $E_G \propto -n^2\gamma^2$ [37], and that $g^{(2)}_\gamma(0) = n^{-2}N^{-1}dE_G(\gamma)/d\gamma$ [90].

In summary, the exact finite-system correlation functions show behaviour consistent with a broad crossover around the mean-field critical value. At stronger interactions, our exact results for small atom numbers are in close agreement with the predictions of mean-field theory.

## 4 Dynamics following an interaction quench

In this section we investigate the nonequilibrium evolution of the attractively interacting Lieb–Liniger gas following an interaction quench for $N = 4$ particles at time $t = 0$. Initially the system is prepared in the ideal-gas ground state, for which the wave function is constant in space, $\psi_0(\{x_i\}) = \langle\{x_i\}|\psi_0\rangle = L^{-N/2}$. Formally, the state of the system at time $t > 0$ is given

by

$$|\psi(t)\rangle = \sum_{\{\lambda_j\}} C_{\{\lambda_j\}}\, e^{-iE_{\{\lambda_j\}}t}|\{\lambda_j\}\rangle\,, \tag{11}$$

where the $C_{\{\lambda_j\}} \equiv \langle\{\lambda_j\}|\psi_0\rangle$ are the overlaps of the initial state with the Lieb–Liniger eigenstates $|\{\lambda_j\}\rangle$ at the postquench interaction strength $\gamma$, and the $E_{\{\lambda_j\}}$ are the corresponding energies. The evolution of equal-time correlation functions (Sec. 2.2) is calculated by noting that the time evolution of the expectation value of an arbitrary operator $\hat{O}$ in the time-dependent state $|\psi(t)\rangle$ is given by

$$\begin{aligned}
\langle\hat{O}(t)\rangle &\equiv \langle\psi(t)|\hat{O}|\psi(t)\rangle \\
&= \sum_{\{\lambda_j\}}\sum_{\{\lambda_j'\}} C^*_{\{\lambda_j'\}} C_{\{\lambda_j\}} e^{i(E_{\{\lambda_j'\}}-E_{\{\lambda_j\}})t}\langle\{\lambda_j'\}|\hat{O}|\{\lambda_j\}\rangle.
\end{aligned} \tag{12}$$

The matrix elements $\langle\{\lambda_j'\}|\hat{O}|\{\lambda_j\}\rangle$ and overlaps $C_{\{\lambda_j\}}$ are calculated with the method described in Ref. [33].

Numerically it is necessary to truncate the infinite sum in Eq. (12), and our truncation procedure is analogous to that described in Appendix A of Ref. [32]: we include all eigenstates for which the populations $|C_{\{\lambda_j\}}|^2$ are larger than some threshold value, thereby minimizing the normalization sum-rule violation $\Delta N = 1 - \sum_{\{\lambda_j\}} |C_{\{\lambda_j\}}|^2$ for the corresponding basis size. For calculations of $\tilde{n}(k_j,t)$ and $g^{(2)}(x,t)$ for interaction-strength quenches to $\gamma = -40$ we use a cutoff $|C_{\{\lambda_j\}}|^2 \geq 10^{-8}$, leading to a sum-rule violation of $\Delta N = 9 \times 10^{-6}$. All other correlation functions are calculated with a more stringent cutoff $|C_{\{\lambda_j\}}|^2 \geq 10^{-10}$, and the sum-rule violations are correspondingly smaller. We have checked that increasing the cutoff does not visibly alter any of our results.

## 4.1 Influence of bound states following a quench

Before investigating the detailed nonequilibrium dynamics of the Lieb–Liniger gas following a quench to attractive interactions, we first consider the populations $|C_{\{\lambda_j\}}|^2$ of the eigenstates of the postquench Hamiltonian, which are constant at all times $t > 0$ [cf. Eq. (11)]. Comparing these populations to those resulting from quenches to repulsive interactions helps provide an understanding of the contribution of bound states to the nonequilibrium dynamics in the attractive case.

In Fig. 3 we plot the populations $|C_{\{\lambda_j\}}|^2$ of several representative Lieb–Liniger eigenstates following quenches of the interaction strength from zero to a wide range of final interaction strengths $\gamma$. [Recall from Sec. 2.1 that for $N = 4$ there are two independent $n_j$ to be specified, which we indicate by the legend in Fig. 3(b).[4]] For attractive interactions [Fig. 3(a)] several eigenstates containing bound states have significant populations for small values of $|\gamma| \lesssim 5$. (Note that the number of particles in the bound state can be inferred from the distribution of the rapidities in the complex plane.) The populations of the ground state $\{n_j\} = \{0,0\}$ (red solid line), which is a four-particle bound state, and the three-particle bound state $\{n_j\} = \{1,0\}$ (green dotted line) are dominant for quenches to $\gamma \gtrsim -4$. However, their populations decrease rapidly with increasing absolute interaction strength beyond $|\gamma| = 4$.

At intermediate interaction strengths $\gamma \simeq -5$, two-body bound states start to dominate the populations [e.g., the states with $\{n_j\} = \{2,0\}$ (blue dot-dashed line) and $\{n_j\} = \{1,1\}$ (pink dot-dot-dashed line)]. For increasingly attractive values of $\gamma$, the populations of gas-like states

---

[4]Note that for repulsive interactions the quantum-number pairs $\{n_j\}$ quoted here refer to the "reduced" quantum numbers, i.e., the excitation numbers relative to the Fermi-sea ground state (cf. Sec. 2.1).

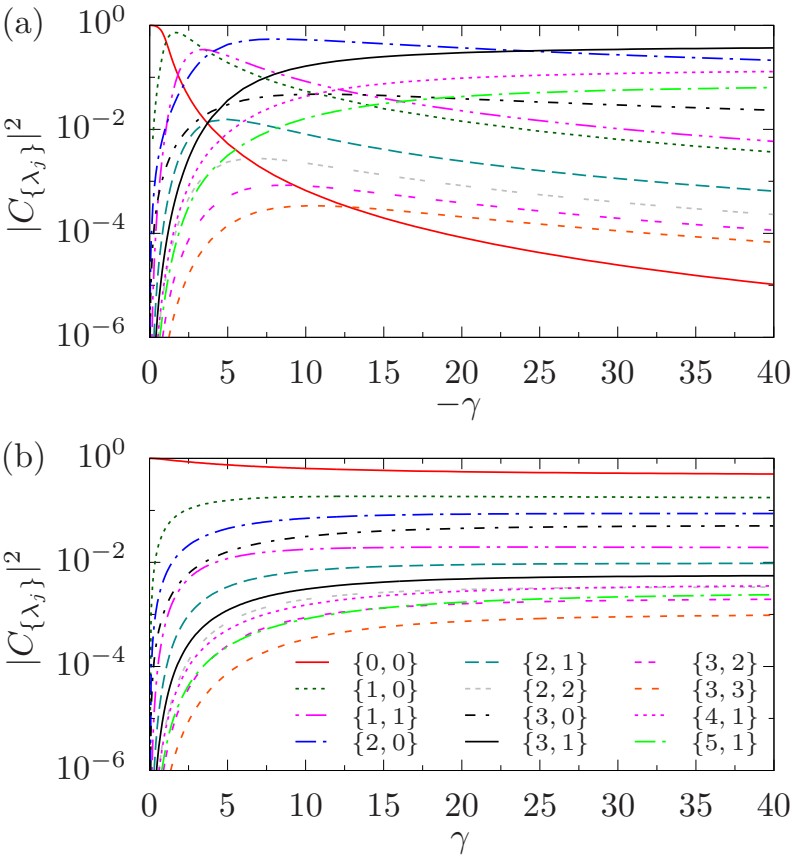

Figure 3: Populations $|C_{\{\lambda_j\}}|^2$ of the lowest-energy Lieb–Liniger eigenstates for quenches of the interaction strength from zero to $\gamma$ and $N = 4$ particles. (a) Populations for attractive postquench interaction strengths. All states except those with $\{n_j\} = \{3,1\}$, $\{4,1\}$, and $\{5,1\}$ contain bound states (i.e., have some complex rapidities). See the detailed discussion of bound states in Sec. 4.1, as well as Fig. 4. (b) Populations for repulsive postquench interaction strengths for comparison with (a).

with no bound-state component grow [e.g., $\{n_j\} = \{3,1\}$ (black solid line) and $\{n_j\} = \{4,1\}$ (pink dotted line)]. Indeed, at $\gamma \simeq -24$, the population of the super-Tonks state $\{n_j\} = \{3,1\}$ — the lowest-energy gas-like state at strong interactions — begins to dominate. However, the two-body bound state with $\{n_j\} = \{2,0\}$ (blue dot-dashed line) still has a significant population in the strongly interacting regime.[5] Consequently, we expect bound states to influence the dynamical evolution of correlation functions following a quench from the ideal gas to all attractive interaction strengths that we consider. Comparing the populations of eigenstates for attractive postquench interactions to those for repulsive interactions, Fig. 3(b), we can see that there is significantly less structure in the populations of the latter states, which are all gas-like. We observe that the populations of excited gas-like eigenstates increase monotonically with increasing $|\gamma|$ for both repulsive and attractive interactions, whereas the results of Fig. 3(a) suggest that the populations of the eigenstates containing bound states all eventually decrease as $\gamma \to -\infty$. We note that although scattering states of the attractive gas connect adiabatically to states of the repulsive gas in the limit $\gamma \to \pm\infty$ [66], the quantum-number labels of the states differ on either side of the infinite-interaction-strength limit. For example, for $N = 4$ particles, the super-Tonks state with $\{n_j\} = \{3,1\}$ connects to the ground state for repulsive

---

[5]We note that at $\gamma = -40$ this state has an energy of $E = -143.9k_F^2$, which is close to the energy of the two-particle McGuire cluster state with $E = -144.1k_F^2$ [37].

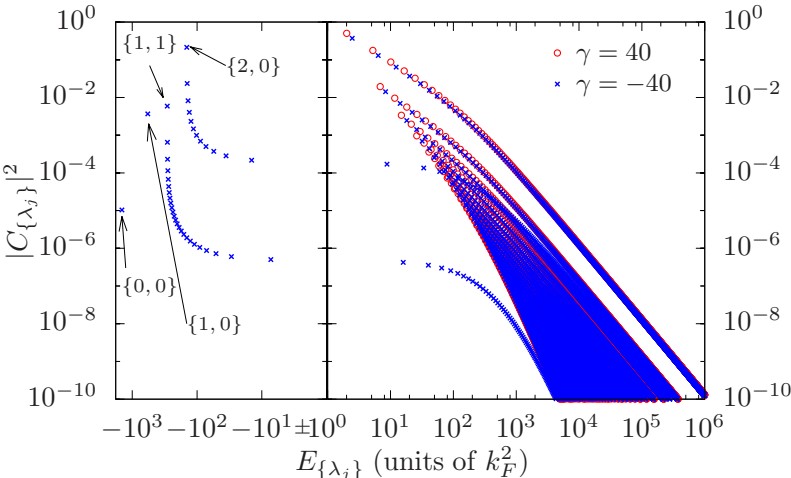

Figure 4: Comparison of populations of eigenstates in the postquench basis for quenches from the ideal-gas ground state to $\gamma = -40$ (blue crosses) and $\gamma = 40$ (red circles) for $N = 4$ particles. To display negative energies on a logarithmic scale, we mirror the energy axis around $E_{\{\lambda_j\}} = 1k_F^2$, plotting the populations of eigenstates with negative energy on the left and those with positive energy on the right. (Note that there are no occupied states with $|E_{\{\lambda_j\}}| < 1k_F^2$.) Four characteristic bound states with negative energy are labelled with their (ideal-gas) quantum numbers $\{n_j\}$, and are described further in the main text.

interactions, $\{n_j\} = \{0, 0\}$.

To better understand the eigenstate contributions to the nonequilibrium dynamics following a quench to attractive interactions, we focus on quenches of $N = 4$ particles from the ideal-gas ground state to attractive and repulsive interactions with $\gamma = \pm 40$, and plot in Fig. 4 the populations $|C_{\{\lambda_j\}}|^2$ of the contributing eigenstates against their energies $E_{\{\lambda_j\}}$. We see that there are additional families of populated states for the attractive gas (sequences of blue crosses that extend to negative energies) that are not present for the repulsive gas (red circles). These are due to four different types of contributing bound states, which we now describe.

The first two types of bound states are four-body and three-body bound states, and each of these types contains only a single populated state. These are, respectively, the ground state $\{n_j\} = \{0, 0\}$ at $E \simeq -1441k_F^2$ with $|C_{0,0}|^2 \simeq 10^{-5}$ and the first parity-invariant excited state $\{n_j\} = \{1, 0\}$ at $E \simeq -576k_F^2$ with $|C_{1,0}|^2 \simeq 3.7 \times 10^{-3}$. We note that the parity invariance of populated eigenstates for quenches from the initial ideal gas [28] restricts the appearance of bound states with more than two bound particles to only these two states.

The third type is represented by the eigenstate with $\{n_j\} = \{2, 0\}$, which has two bound particles and two free particles, and is the first in a family of similar states $\{2+l, 0\}$ ($l$ a nonnegative integer) whose populations decrease monotonically with increasing $l$. The fourth type is represented by the eigenstate with $\{n_j\} = \{1, 1\}$, which contains two two-particle bound states, and is the first in a family with decreasing populations for higher excitations which alternate between the quantum numbers $\{1 + l, 1 + l\}$ and $\{1 + l, l\}$, with $l$ a positive integer. For larger $l$, the two two-body bound states have higher "centre-of-mass" momenta with opposite sign (recall that only eigenstates with total momentum $P = 0$ have nonzero occupations following the quench), and for $l > 12$ the corresponding positive centre-of-mass energy of the pairs exceeds the sum of their binding energies.

We can see from Fig. 4 that the distributions of populations over gas-like eigenstates are similar for quenches to $\gamma = \pm 40$, aside from a shift in energy and a small decrease in populations for the attractive gas due to the appearance of the additional bound states. In par-

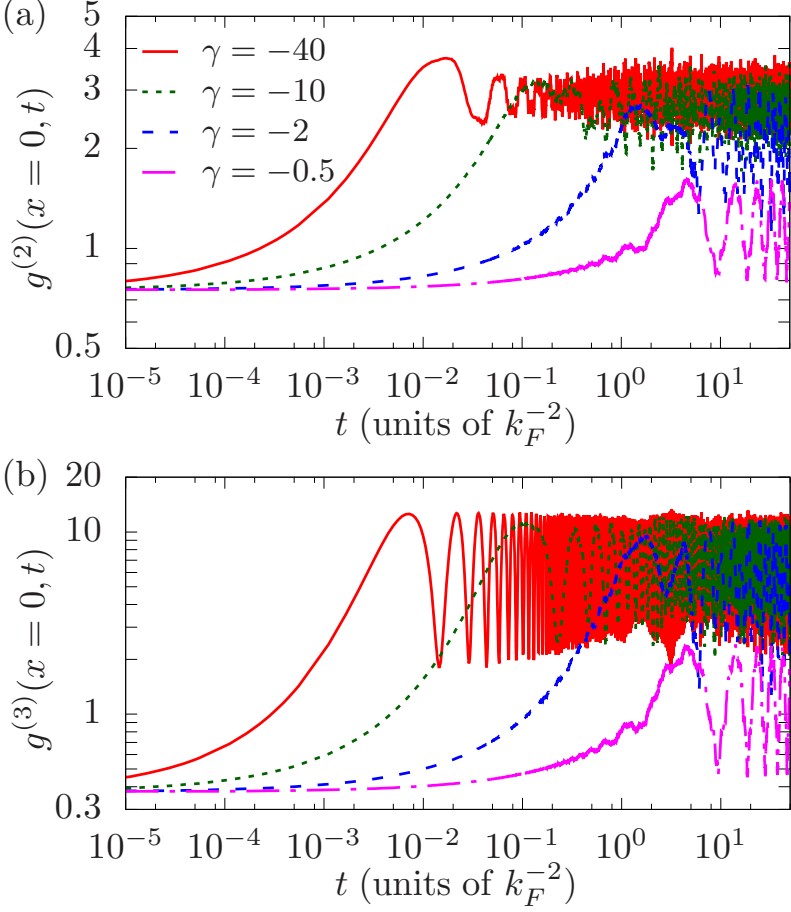

Figure 5: Time evolution of local correlation functions following quenches of the interaction strength from zero to $\gamma = -0.5, -2, -10$ and $-40$ for $N = 4$ particles. (a) Local second-order correlation $g^{(2)}(x = 0, t)$. (b) Local third-order correlation $g^{(3)}(x = 0, t)$.

ticular, the number of eigenstates with populations $|C_{\{\lambda_j\}}|^2 \geq 10^{-10}$ is 7815 (7462) for the attractive (repulsive) gas. The shift in energy can be explained by noting that for $\gamma = \pm 40$, the system is in the strongly interacting regime and the Bethe rapidities of scattering states (i.e. states with no bound particles) can be obtained by a strong-coupling expansion around the Tonks–Girardeau limit of infinitely strong interactions (see, e.g., Ref. [91]). This yields $\lambda_j \simeq (1 - 2/\gamma)k_j$, where the $k_j$ are the Tonks–Girardeau values, implying opposite energy shifts in the attractive and repulsive cases.

## 4.2 Dynamics of local correlations

We now consider the nonequilibrium dynamics following the quench. In Fig. 5(a) we plot the local second-order correlation $g^{(2)}(x = 0, t)$ for $N = 4$ particles following a quench from $\gamma = 0$ to four representative final interaction strengths. Initially, $g^{(2)}(0, t = 0) = 1 - 1/N = 0.75$ (cf. Sec. 3.1). For a quench to $\gamma = -0.5$ (pink dot-dashed line), $g^{(2)}(0, t)$ shows nearly monochromatic oscillatory behaviour. This is similar to the behaviour following quenches to small repulsive interaction strengths analysed in Ref. [32]. Because the difference between the postquench energy $E \equiv \langle \psi(0^+)|\hat{H}|\psi(0^+)\rangle = (N-1)n^2\gamma$ [32, 92] and the ground-state energy of the system is small compared to the finite-size energy gap to the first (parity-invariant) excited state, the ensuing dynamics are dominated by these two states, and the energy difference between them determines the dominant frequency of the oscillations.

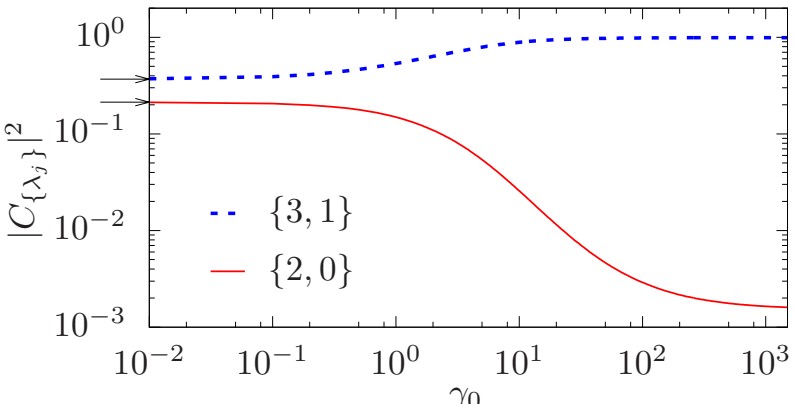

Figure 6: Populations $|C_{\{\lambda_j\}}|^2$ of the super-Tonks state $\{n_j\} = \{3,1\}$ and the dominant two-body bound state $\{n_j\} = \{2,0\}$ (see text) for quenches from the interacting ground state at $\gamma_0 > 0$ to $\gamma = -40$ for $N = 4$ particles. The black arrows indicate the populations for the quench from the ideal-gas ground state.

Quenches to more attractive values of $\gamma$ show the generic behaviour of an initially rising $g^{(2)}(0,t)$ that eventually fluctuates about a seemingly well-defined average value. The frequencies of the oscillations are determined by the energy differences between the Lieb–Liniger eigenstates with the largest populations. For example, for $\gamma = -40$ (solid red line), the postquench wave function is dominated by the super-Tonks state and the first two-body bound state, cf. Fig. 3, and the dominant frequency in the oscillations at early times matches the energy difference between these two eigenstates. At later times, the shape of $g^{(2)}(0,t)$ is more irregular, but the large oscillations due to the two dominant eigenstates persist.

In Fig. 5(b) we plot the local third-order correlation $g^{(3)}(x = 0, t)$ for $N = 4$ particles following a quench from $\gamma = 0$ to the same four final interaction strengths. Initially, $g^{(3)}(0, t = 0) = N(N-1)(N-2)N^{-3} = 0.375$ (see Sec. 3.2). For small postquench interaction strengths, $\gamma = -0.5$ (pink dot-dashed line) and $\gamma = -2$ (blue dashed line), the evolution is qualitatively similar to that of $g^{(2)}(x = 0, t)$ for the same interaction strengths. For larger attractive values of the postquench interaction strength, on the other hand, the shape of $g^{(3)}(x = 0, t)$ is more regular compared to $g^{(2)}(x = 0, t)$, reflecting the fact that only one three-body bound state contributes to the postquench wave function, whereas multiple states containing bound pairs are present. Indeed for $\gamma = -10$ (green dotted line) and $\gamma = -40$ (solid red line), $g^{(3)}(0, t)$ is dominated by a single frequency, given by the energy difference between the three-body bound state $\{n_j\} = \{1,0\}$ and the predominant two-body bound state $\{n_j\} = \{2,0\}$. The initial rise of both $g^{(2)}(0, t)$ and $g^{(3)}(0, t)$ terminates on an increasingly shorter time scale with increasingly attractive postquench interaction strength. This time scale corresponds to about half the period of the ensuing oscillations and is proportional to $\gamma^{-2}$, corresponding to the scaling of the energy $E_{\{\lambda_j\}} \propto -\gamma^2$ of eigenstates containing bound states [37].

For quenches from the ideal-gas initial state, we find that the population of the bound states leads to significantly increased values of both $g^{(2)}(0, t)$ and $g^{(3)}(0, t)$ — in stark contrast to the decay of the same quantities following quenches to repulsive interactions [32] due to the "fermionization" of the system. Such large values of these local correlation functions would lead to strong particle losses in experiments [7, 93, 94]. This is in contrast to the observations in the quench experiments performed in Ref. [7], where the quasi-one-dimensional gas was quenched from strongly repulsive interactions to strongly attractive interactions, and no significant losses were observed. In such a scenario the overlap of the initial strongly repulsive

ground state with the super-Tonks state is dominant, and the bound states thus acquire only small populations in the course of the quench [7, 63, 64, 66].

To investigate the influence of the initial state on the populations of the two most dominant postquench eigenstates (cf. Fig. 3), we find the (correlated) ground state $|\psi_0\rangle$ of the system at $\gamma_0 > 0$ and then compute the populations of the eigenstates following a quench to $\gamma = -40$. In Fig. 6, we plot the populations $|C_{2,0}|^2$ and $|C_{3,1}|^2$ of the aforementioned two-body bound state and the super-Tonks state, respectively, for a wide range of initial values $\gamma_0$. Starting in the strongly interacting regime $\gamma_0 = 10^3$, the overlap between the initial (Tonks–Girardeau) state and the super-Tonks state is close to unity. As $\gamma_0$ is decreased, the population of the super-Tonks gas decreases, while the population of the bound state increases. At $\gamma_0 \simeq 1$, the two populations are already near their respective values following a quench from the ideal-gas initial state (indicated by black arrows on the left-hand side). The results of Fig. 6 suggest that the postquench values of $g^{(2)}(0, t)$ and $g^{(3)}(0, t)$ would be much smaller for quenches from initial values of $\gamma_0 \gtrsim 10$ compared to those from the noninteracting initial state.

## 4.3 Dynamics of the momentum distribution

We now turn our attention to the postquench dynamics of the momentum distribution. Quenches from the ideal-gas ground state with $N = 4$ particles to three different values of $\gamma$ are compared in Fig. 7. In each case we plot the time evolution of the momentum-mode occupations $\widetilde{n}(k_j, t)$ [cf. Eq. (8)] for the first six nonnegative momentum modes $k_j$ ($j = 0, 1, \ldots, 5$). Initially, all particles occupy the zero-momentum single-particle orbital, $\widetilde{n}(k_j, t = 0) = N\delta_{j0}$. At times $t > 0$, the interaction quench leads to a redistribution of this population over other single-particle modes. At early times, all nonzero modes rise with the same rate, independent of $k$, due to the local nature of the interaction potential, which corresponds to a momentum-independent coupling [95]. This applies to all postquench interaction strengths $\gamma$, but the time at which deviations from this behaviour first appear depends on $\gamma$.

All quenches show the same generic behaviour — the momentum-mode populations eventually level off and fluctuate about a well-defined value. These populations undergo oscillations with frequencies determined by the energy differences between the dominant Lieb–Liniger eigenstates. For example, for the $\gamma = -40$ case of Fig. 7(c) each mode exhibits fast oscillations at a single frequency given by the energy difference between the super-Tonks state $\{n_j\} = \{3, 1\}$ and the two-body bound state $\{n_j\} = \{2, 0\}$, superposed with some irregular envelope function.

In Fig. 8, we compare $\widetilde{n}(k = 0, t)$ for quenches from the ideal gas to repulsive and attractive interaction strengths of the same magnitude. In Fig. 8(a), we plot the time evolution of the zero-momentum mode occupation $\widetilde{n}(0, t)$ for quenches from $\gamma = 0$ to $\gamma = -10$ (solid red line) and $\gamma = 10$ (blue dashed line). The envelope of $\widetilde{n}(0, t)$ for attractive interactions is similar to the shape of $\widetilde{n}(0, t)$ for repulsive interactions. For the quench to attractive interactions, $\widetilde{n}(0, t)$ shows large regular oscillations on top of this envelope. This also applies for quenches to $\gamma = \pm40$, Fig. 8(b), but the oscillations for quenches to $\gamma = -40$ (solid red line) are faster than for quenches to $\gamma = -10$. The correspondence between $\widetilde{n}(0, t)$ following a quench to strong attractive interactions and that following a quench to equally strong repulsive interactions reflects the fact that the two postquench wave functions are similar in their composition, aside from the additional presence of two-body bound states for attractive interactions, as illustrated in Fig. 4.

We also observe a partial revival in $\widetilde{n}(0, t)$ for quenches to $\gamma = \pm40$. This revival is due to the proximity of the system at $\gamma = 40$ to the Tonks–Girardeau limit of infinitely strong interactions, where the spectrum of the repulsive Lieb–Liniger model is identical to that of free fermions [96]. This also applies to the scattering states of the attractive system. For $\gamma = \pm\infty$, this would lead to recurrences at integer multiples of $t_{\mathrm{rev}} = 3.5k_F^{-2}$ [32] due to the commen-

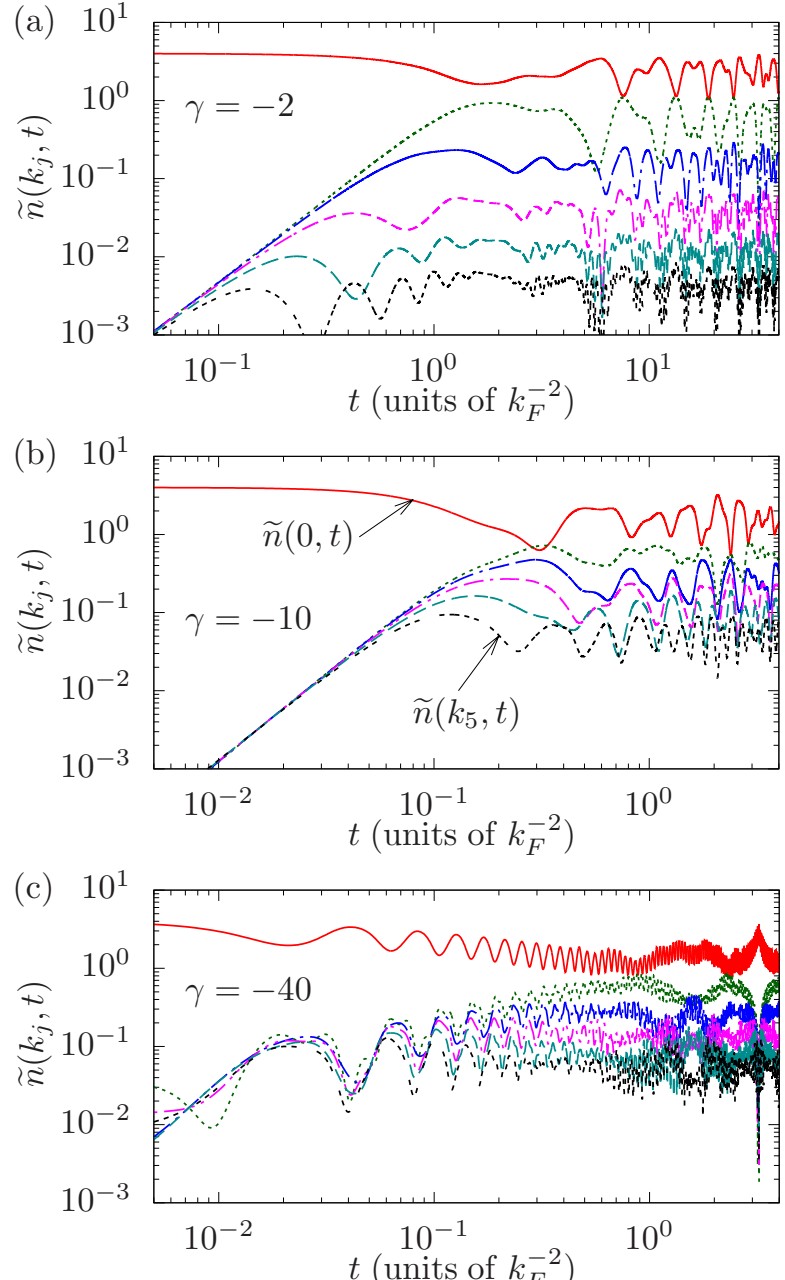

Figure 7: Time evolution of the momentum occupations $\widetilde{n}(k_j, t)$ of the first six nonnegative momentum modes $k_j$ ($j = 0, 1, \ldots, 5$) for $N = 4$ particles and for a quench of the interaction strength from zero to (a) $\gamma = -2$, (b) $\gamma = -10$, and (c) $\gamma = -40$. Note the different range of the time axis of (a) compared to that of (b) and (c).

surability of eigenstate energies [97]. However, for the finite interaction strengths considered here, the revival time is shifted to a later time $t_{\mathrm{rev}} \simeq 3.9 k_F^{-2}$ for repulsive interactions [32] and to an earlier time $t_{\mathrm{rev}} \simeq 3.2 k_F^{-2}$ for attractive interactions, due to the finite-coupling corrections to the Bethe rapidities discussed in Sec. 4.1.

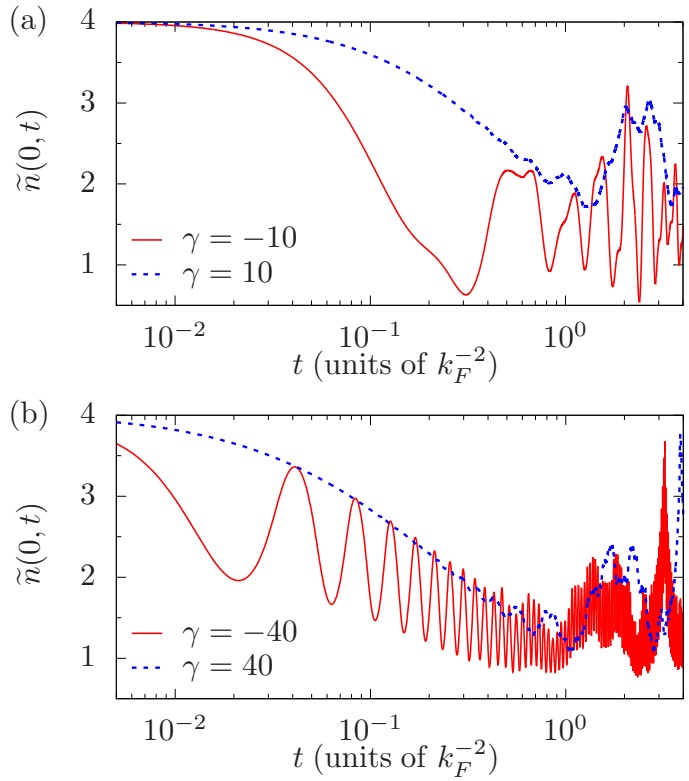

Figure 8: Time evolution of the zero-momentum mode occupation $\tilde{n}(0,t)$ for $N = 4$ particles and quenches of the interaction strength from zero to attractive and repulsive values of the same magnitude. (a) Post-quench interaction strengths of $\gamma = -10$ (red solid line) and $\gamma = 10$ (blue dashed line). (b) Post-quench interaction strengths of $\gamma = -40$ (red solid line) and $\gamma = 40$ (blue dashed line).

## 4.4   Dynamics of nonlocal pair correlations

We now consider the evolution of the full nonlocal second-order correlation $g^{(2)}(x,t)$. In Fig. 9 we plot the behaviour of this quantity for an interaction quench from zero to $\gamma = -40$ for $N = 4$ particles. Figure 9(a) shows $g^{(2)}(x,t)$ at four representative times $t$. Initially, $g^{(2)}(x,0) = 1 - 1/N$ (horizontal line). At $t = 0.01 k_F^{-2}$ (red dashed line), the local value is already greatly enhanced, $g^{(2)}(0, t = 0.01 k_F^{-2}) \simeq 3.5$, cf. Fig. 5(a). [The scale of the $y$-axis is chosen so that the long-range features of $g^{(2)}(x)$ are visible, and the large values for $x \lesssim 0.02 \times (2\pi k_F^{-1})$ are therefore cut off.] In addition to the central peak, at separations $x \simeq 0.1 \times (2\pi k_F^{-1})$ a secondary peak emerges, while at larger distances $g^{(2)}(x)$ exhibits a decaying oscillatory structure. As time progresses, this secondary peak propagates away from the origin and broadens as can be seen at, e.g. $t = 0.1 k_F^{-2}$ (green dotted line) and $t = 0.25 k_F^{-2}$ (blue dot-dashed line).

The build-up of this secondary correlation peak and its propagation through the system can be more clearly seen in Fig. 9(b), where we plot the time-evolution of $g^{(2)}(x,t)$ up to $t = 0.25 k_F^{-2}$. The propagation of this peak is consistent with $x(t) \propto t^{1/2}$, which was also observed for quenches from the same initial state to strongly repulsive interactions [29, 32]. (Note that the colour scale is chosen so that the long-range behaviour is visible, and the local second-order correlation is again not resolved.) Figure 9(c) shows $g^{(2)}(x,t)$ for longer times up to $t = 4 k_F^{-2}$. The overall structure on this longer time scale is more complicated, with several soliton-like correlation dips propagating through the system [32] and a partial revival

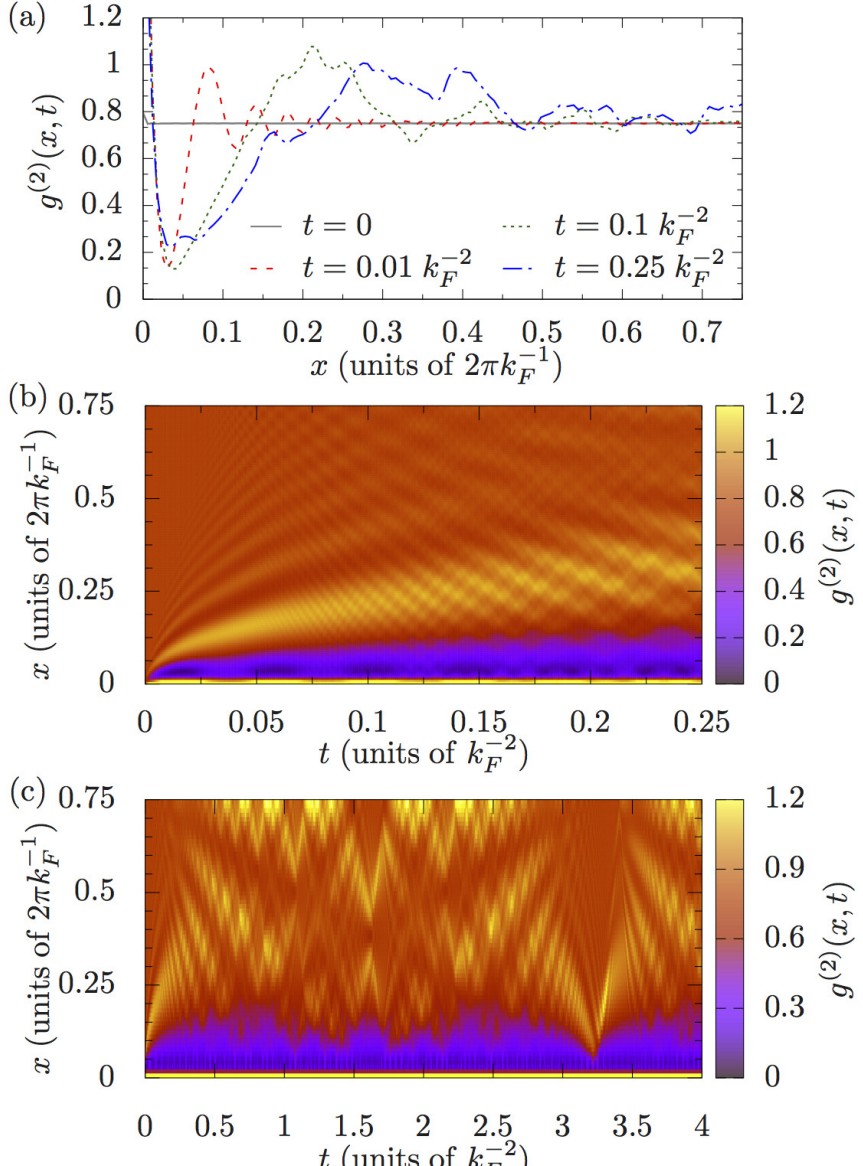

Figure 9: Time evolution of the nonlocal second-order correlation function $g^{(2)}(x,t)$ following a quench from the ideal-gas ground state to $\gamma = -40$ for $N = 4$ particles. (a) Correlation function $g^{(2)}(x)$ at four representative times. (b) Evolution of $g^{(2)}(x,t)$ for short times $t \leq 0.25\,k_F^{-2}$ and (c) longer times $t \leq 4\,k_F^{-2}$. Note that the colour scale has been chosen so as to preserve the visibility of long-range features, and thus $g^{(2)}(x,t)$ for $x \lesssim 0.02 \times (2\pi k_F^{-1})$ is not resolved. The local value oscillates between $g^{(2)}(0,t) \simeq 2$ and $\simeq 4$, cf. Sec. 4.2.

of $g^{(2)}(x, t = 0)$ at $t \simeq 3.2 k_F^{-2}$ [cf. Figs. 7(c) and 8(b)]. Besides the significantly increased value at small distances, the behaviour of $g^{(2)}(x,t)$ is strikingly similar to the results obtained in Ref. [32] for quenches from the same noninteracting ground state to repulsive final interaction strengths.

In summary, quenches from the ideal-gas ground state to attractive values of $\gamma$ result in the occupation of energy eigenstates containing bound states in addition to the gas-like scattering states of the attractively interacting model, which are analogous to the eigenstates of the repulsively interacting Lieb–Liniger gas. As the magnitude $|\gamma|$ of the final interaction strength is increased, the postquench occupations of the gas-like excited states approach those of their

counterparts following a quench to the corresponding repulsive interaction strength, and the occupations of bound states eventually decrease. However, these bound states significantly influence the dynamics of postquench correlation functions for all final interaction strengths we have considered, causing large oscillations in local correlations and in the occupation of the zero-momentum mode. For large attractive values of $\gamma$, bound states are highly localized and thus influence the second-order correlation function only at small separations, whereas at larger separations this function exhibits postquench dynamics similar to those observed following quenches to repulsive interactions [32].

# 5 Time-averaged correlations

A closed quantum-mechanical system prepared in a pure state will remain in a pure state for all time. However, for a nondegenerate postquench energy spectrum, as is the case here (cf. Refs. [32, 33]), the energy eigenstates will dephase, and the time-averaged expectation value of any operator $\hat{O}$ can be expressed in terms of its diagonal matrix elements between energy eigenstates,

$$
\begin{aligned}
\langle \hat{O} \rangle_{\text{DE}} &= \lim_{\tau \to \infty} \frac{1}{\tau} \int_0^\tau dt \, \langle \psi(t) | \hat{O} | \psi(t) \rangle \\
&= \sum_{\{\lambda_j\}} |C_{\{\lambda_j\}}|^2 \langle \{\lambda_j\} | \hat{O} | \{\lambda_j\} \rangle .
\end{aligned}
\tag{13}
$$

This quantity can be viewed as the expectation value of $\hat{O}$ in the diagonal-ensemble density matrix [73]

$$
\hat{\rho}_{\text{DE}} = \sum_{\{\lambda_j\}} |C_{\{\lambda_j\}}|^2 |\{\lambda_j\}\rangle \langle \{\lambda_j\}| .
\tag{14}
$$

We note that in practice the sum in Eq. (14) runs over a finite set of energy eigenstates with populations $|C_{\{\lambda_j\}}|^2$ exceeding some threshold value. If the expectation value of an operator relaxes at all, it must relax to the corresponding diagonal-ensemble value [98]. Although expectation values may exhibit rather large fluctuations around their time-averaged values for system sizes as small as those considered here, in general the relative magnitude of these fluctuations should decrease with increasing system size and vanish in the thermodynamic limit. However, establishing this behaviour is beyond the scope of the current work and we will simply regard the diagonal ensemble defined by Eq. (14) as the ensemble appropriate to describe the relaxed state of the finite-sized system. In the following we consider the time-averaged properties of the quenched system.

## 5.1 Local correlations

In Fig. 10(a), we plot the enhancement of the diagonal-ensemble value $g_{\text{DE}}^{(2)}(0)$ of the local second-order correlation over the initial noninteracting value $g_{\gamma=0}^{(2)}(0)$ of this function following an interaction quench from zero to $\gamma$ for particle numbers $N = 2$, 3, and 4. For all particle numbers $N$ considered, as $|\gamma|$ is increased from the ideal-gas limit, $g_{\text{DE}}^{(2)}(0)$ initially increases rapidly before reaching a local maximum, which occurs at smaller values of $|\gamma|$ for larger particle numbers $N$. For $N = 4$ particles (solid blue line) this local maximum in $g_{\text{DE}}^{(2)}(0)$ occurs at $\gamma = -1$ and coincides with the crossing of the population of the three-particle bound state $\{n_j\} = \{1, 0\}$ and that of the ground state [see Fig. 3(a)]. The local minimum of $g_{\text{DE}}^{(2)}(0)$ at $\gamma = -1.5$ coincides with the maximum population of this three-particle bound state, and as

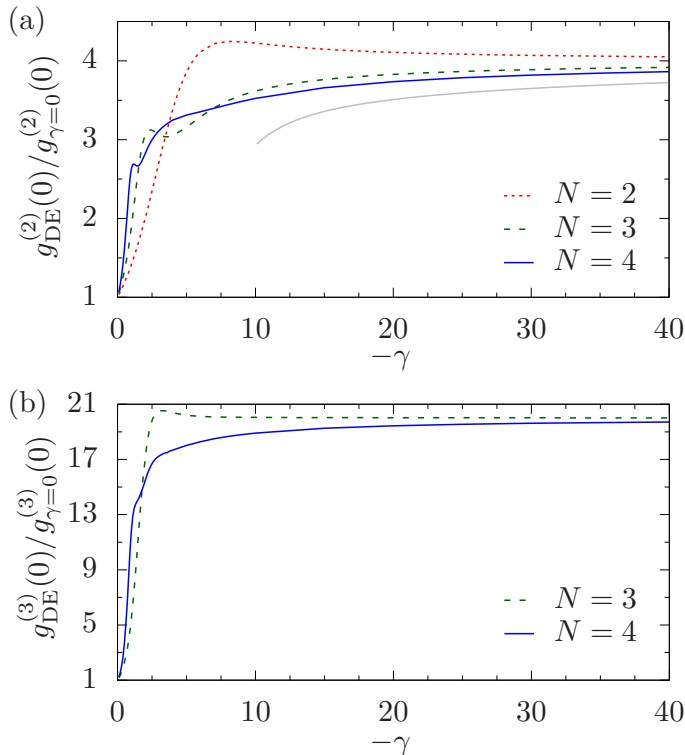

Figure 10: Diagonal-ensemble values of local correlation functions following quenches of the interaction strength from zero to $\gamma$. (a) Enhancement $g^{(2)}_{\mathrm{DE}}(0)/g^{(2)}_{\gamma=0}(0)$ of the local second-order correlation over the initial ideal-gas value, for quenches to $\gamma$ for particle numbers $N = 2, 3$, and 4. The light grey solid line indicates the quench-action strong-coupling (order-$1/\gamma^3$) thermodynamic-limit prediction for the stationary value of $g^{(2)}(0)$ [35, 36]. (b) Enhancement $g^{(3)}_{\mathrm{DE}}(0)/g^{(3)}_{\gamma=0}(0)$ of the local third-order correlation over the ideal-gas value, for quenches to $\gamma$ and particle numbers $N = 3$ and 4.

soon as the population of this state starts to decrease, $g^{(2)}_{\mathrm{DE}}(0)$ begins to increase monotonically with increasing $|\gamma|$.

For large attractive values of $\gamma$, the local second-order correlation appears to tend to a constant value $g^{(2)}_{\mathrm{DE}}(0)/g^{(2)}_{\gamma=0}(0) \simeq 4$, which is much larger than the ideal gas and super-Tonks values [67]. The decrease of $g^{(2)}_{\mathrm{DE}}(0)$ with increasing particle number at fixed large $|\gamma|$ appears consistent with an approach towards the quench-action thermodynamic-limit strong-coupling value obtained to third order in $1/\gamma$ in Refs. [35, 36], indicated by the solid grey line, as $N \to \infty$.

Using the quench-action approach [71, 72] in the thermodynamic limit, Refs. [35, 36] found that $g^{(2)}_{\mathrm{DE}}(0) = 2$ for $\gamma \to 0^-$. Our calculations do not recover this result for small values of $|\gamma|$, as our small system sizes lead to a finite-size gap for excitations and therefore the energy added by the quench is small in this case. Additionally, eigenstates with more than four bound particles are trivially absent in our calculations, whereas for small postquench values of $|\gamma|$ they contribute significantly in the analysis of Refs. [35, 36]. For larger values of $|\gamma|$, however, states with more than two bound particles are strongly suppressed and we expect our results to be less influenced by finite-size effects [33].

In Fig. 10(b), we plot the enhancement of the diagonal-ensemble value of the local third-order correlation $g^{(3)}_{\mathrm{DE}}(0)$ over its noninteracting initial value following an interaction quench from zero to $\gamma$ for particle numbers $N = 3$ and 4. The qualitative behaviour is similar to that

of $g_{\mathrm{DE}}^{(2)}(0)$. For strong interactions, $g_{\mathrm{DE}}^{(3)}(0)$ also appears to tend to a constant value that is much larger than the initial value. Whether this result persists for larger atom numbers is an important open question, given that large values of $g^{(3)}(0)$ lead to strong recombination losses in experiments with ultracold gases [93, 94].

## 5.2 Nonlocal correlations

In Fig. 11(a) we plot the momentum distribution $\widetilde{n}_{\mathrm{DE}}(k)$ in the diagonal ensemble for $N = 4$ particles and for several postquench interaction strengths $\gamma$. At high momenta and for all interaction strengths $\gamma$, $\widetilde{n}_{\mathrm{DE}}(k)$ exhibits a scaling of $\widetilde{n}_{\mathrm{DE}}(k) \propto k^{-4}$. This behaviour is due to the universal character of short-range two-body interactions [83–85]. For $\gamma = -0.5$ (pink squares), the functional form of $\widetilde{n}_{\mathrm{DE}}(k)$ is nearly perfectly given by this $\propto k^{-4}$ scaling, and only the three lowest resolvable nonzero momentum modes in our finite periodic system deviate slightly from it.

For a quench to $\gamma = -2$ (blue filled circles), the low-momentum part of $\widetilde{n}_{\mathrm{DE}}(k)$ starts to deviate more strongly from the $\propto k^{-4}$ scaling, and the distribution seems to get wider at low momenta. This low-$k$ "hump" broadens with increasing postquench interaction strength. This behaviour is qualitatively similar to our earlier results for quenches to repulsive values of $\gamma$, where an infrared scaling of $\widetilde{n}_{\mathrm{DE}}(k) \propto k^{-2}$ extends to larger values of $k$ with increasing $\gamma$ [32], consistent with the dependence of the populations $|C_{\{\lambda_j\}}|^2$ on the rapidities $\{\lambda_j\}$ and with analytic results for the postquench momentum distribution in the limit of a quench to infinitely strong repulsive interactions [29]. From the results presented in Fig. 11(a) it is unclear if the emerging hump in the present case of quenches to attractive interactions is consistent with $\propto k^{-2}$ scaling.

In Fig. 11(b), we plot the second-order correlation function $g_{\mathrm{DE}}^{(2)}(x)$ in the diagonal ensemble for the same postquench interaction strengths $\gamma$ as in Fig. 11(a) and compare these to the initial-state form $g^{(2)}(x, t = 0) = 1 - 1/N$ of this function (horizontal line). The first feature we notice is that for all values of the postquench interaction strength, $g_{\mathrm{DE}}^{(2)}(x)$ is increased at small separations $x$ compared to its initial value [cf. Fig. 10(a)]. For the quench to $\gamma = -0.5$ (pink dot-dashed line), $g_{\mathrm{DE}}^{(2)}(x)$ decreases monotonically with increasing $x$. [Due to the periodic nature of our geometry, correlation functions are symmetric around $x = L/2$, and we therefore only show $g_{\mathrm{DE}}^{(2)}(x)$ up to this point.] For $\gamma = -2$ (blue dashed line), $g_{\mathrm{DE}}^{(2)}(x)$ exhibits a local minimum at a finite separation $x \simeq 0.3 \times (2\pi k_F^{-1})$, before increasing again at larger separations. This behaviour can also be observed for $\gamma = -10$ (green dotted line), where the minimum in $g_{\mathrm{DE}}^{(2)}(x)$ moves to smaller separations $x \simeq 0.1 \times (2\pi k_F^{-1})$ and becomes more pronounced. For $\gamma = -40$ (solid red line), the minimum is located at $x \simeq 0.03 \times (2\pi k_F^{-1})$ and its magnitude is again decreased compared to the quench to $\gamma = -10$. We note that the increase of $g_{\mathrm{DE}}^{(2)}(x)$ for $x \gtrsim 0.6 \times (2\pi k_F^{-1})$ is a finite-size effect (cf. Ref. [32]).

In Fig. 11(c), we compare $g_{\mathrm{DE}}^{(2)}(x)$ following a quench to $\gamma = -40$ (red solid line) to that following a quench to $\gamma = 40$ (black dot-dashed line). The shape of $g_{\mathrm{DE}}^{(2)}(x)$ for interparticle separations $x \gtrsim 0.05 \times (2\pi k_F^{-1})$ is similar for both quenches. The main difference is in the short-range behaviour, which is significantly influenced by the highly localized bound states for the quench to attractive interactions. For the quench considered here, the dominant bound-states are two-particle clusters (cf. Fig. 3). In Fig. 11(c) we plot the matrix element $\langle \{\lambda_j\}|\hat{g}^{(2)}(x)|\{\lambda_j\}\rangle$ of the two-body correlation function in the dominant two-body bound state $\{n_j\} = \{2, 0\}$ (blue dashed line). For $N = 2$ particles, the wave function of such a bound state $\Psi(x_1, x_2) \propto \exp(-|x_1 - x_2|/a_{1\mathrm{D}}) = \exp(-|x_1 - x_2|n\gamma/2)$ [64], where $a_{1\mathrm{D}}$ is the 1D scattering length [20, 60]. This implies a two-body correlation $g^{(2)}(x) \propto |\Psi(0, x)|^2 = \exp(-xn\gamma)$, which is indeed consistent with the form of $g^{(2)}(x)$ in the state $\{n_j\} = \{2, 0\}$ at small separations, whereas at larger separations $g^{(2)}(x)$ in this state tends to a constant finite value,

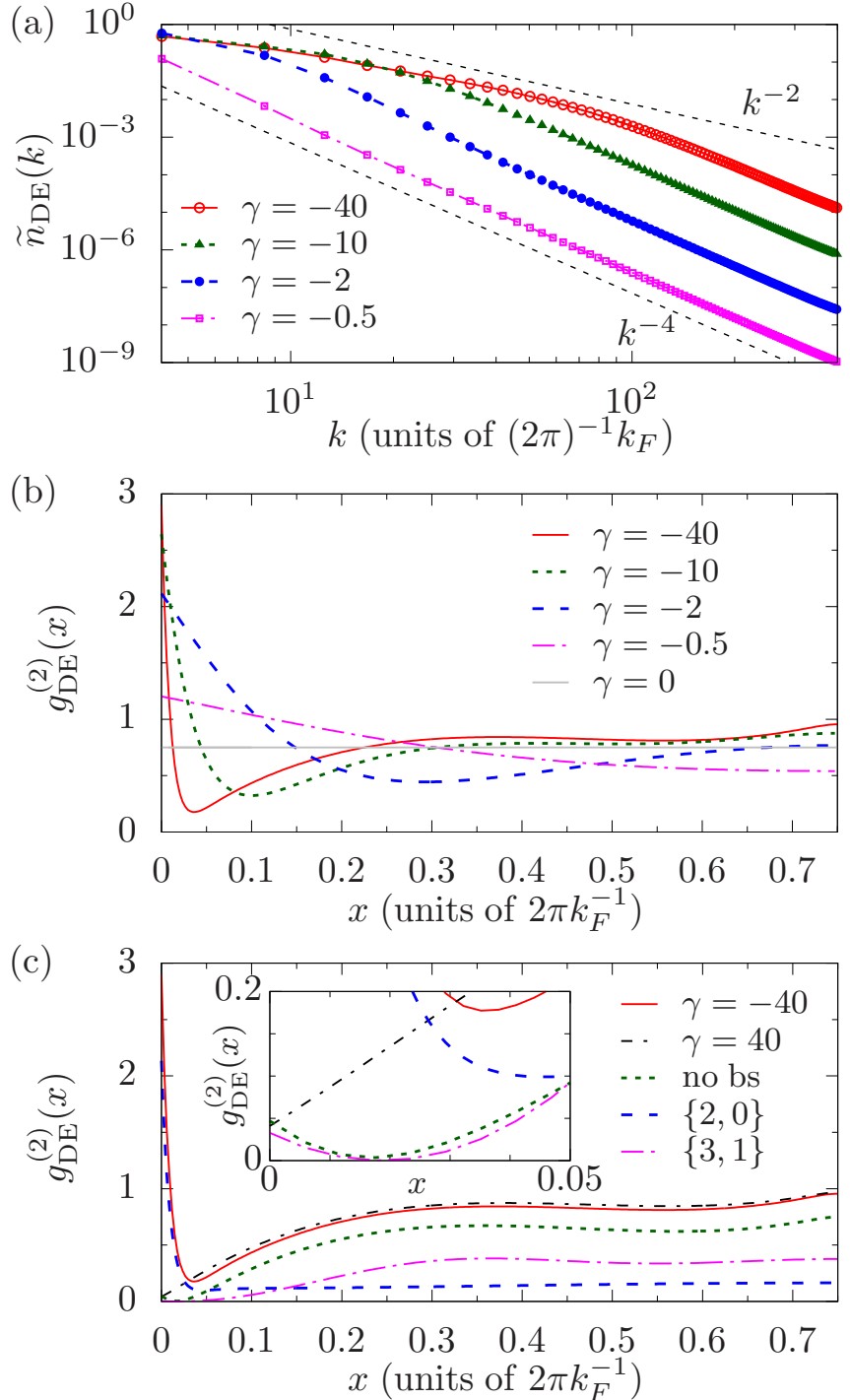

Figure 11: Diagonal-ensemble correlation functions for quenches to $\gamma = -0.5, -2, -10$, and $-40$ for $N = 4$ particles. (a) Momentum distribution $\widetilde{n}_{\mathrm{DE}}(k)$. Black dashed lines indicate scalings $\propto k^{-2}$ (upper line) and $\propto k^{-4}$ (lower line). (b) Second-order correlation $g_{\mathrm{DE}}^{(2)}(x)$. The grey horizontal line indicates the initial value $g^{(2)}(x, t = 0)$. (c) Matrix elements $\langle\{\lambda_j\}|\hat{g}^{(2)}(x)|\{\lambda_j\}\rangle$ of the second-order correlation in representative eigenstates. The inset shows these correlations at small separations $x$, with the result for the super-Tonks state $\{n_j\} = \{3, 1\}$ (pink dot-dashed line) scaled by a factor of 10 for visibility.

due to the unbound particles it contains. Away from small separations, a small proportion of $g_{\text{DE}}^{(2)}(x)$ is due to such contributions of free particles in eigenstates containing bound particles, but this function is dominated by the contributions of scattering states. For attractive interactions these scattering states are expected to be identical to states of the one-dimensional Bose gas with hard-sphere interactions outside the corresponding hard-sphere radius $a_{\text{hs}} \simeq a_{1D} = -2(\gamma n)^{-1} = 0.01875 \times (2\pi k_F^{-1})$ [64]. Indeed from the inset to Fig. 11(c) we observe that the form of $g^{(2)}(x)$ in the super-Tonks state $\{n_j\} = \{3, 1\}$ (pink dot-dashed line, multiplied by a factor of 10 for visibility) and that of $g_{\text{DE}}^{(2)}$ following a quench to $\gamma = -40$ without the contribution of bound states (green dotted line) are broadly consistent with this expectation.

In summary, our results for the time-averaged local second-order correlation function $g_{\text{DE}}^{(2)}(0)$ are consistent with an enhancement of this quantity over the initial ideal-gas value by a factor of $\simeq 4$ in the limit of strong final interaction strengths, and thus with the predictions of Refs. [35, 36] in this limit. Our calculations also reveal an enhancement of the local third-order correlation function $g_{\text{DE}}^{(3)}(0)$ over the ideal-gas value by a factor of $\simeq 20$ for strong interactions, suggesting that the postquench state would be susceptible to large three-body recombination losses in practice. Results for time-averaged correlation functions at interparticle separations larger than the characteristic extent of bound states are comparable to those obtained previously [32] for quenches to repulsive interactions.

# 6  Conclusions

We have studied the nonequilibrium dynamics of the one-dimensional Bose gas following a quantum quench from the noninteracting ground state to attractive interaction strengths $\gamma < 0$. In particular we calculated equilibrium, nonequilibrium, and time-averaged correlation functions of the system and investigated their dependence on the final interaction strength. To achieve this we extended a previously developed coordinate Bethe ansatz method for the nonequilibrium dynamics of the Lieb–Liniger model [33] to the attractively interacting regime. Compared with the case of repulsive interactions, the computational evaluation is found to be significantly more demanding. This is a consequence of near cancellations in the scattering factors of Bethe ansatz wave functions for strongly negative interaction strengths.

We calculated first-, second-, and third-order correlation functions of the ground state for up to seven particles and a wide range of negative interaction strengths $\gamma$, and observed the emergence of bright-soliton-like correlations. As the interaction strength $\gamma$ becomes more negative, the correlation functions approach a form corresponding to bright-soliton solutions of the mean-field approximation.

We then calculated the nonequilibrium correlation functions of a system of four particles following quenches of the interaction strength from $\gamma = 0$ to several different values of $\gamma < 0$. For a small postquench interaction strength $\gamma = -0.5$, the excitation energy imparted to the system by the quench is of the order of the finite-size energy gap, and consequently excitations are strongly suppressed. This results in correlation functions exhibiting quasi-two-level dynamics. For quenches to intermediate attractive values of the interaction strength, the local correlations are found to increase on short time scales and at later times fluctuate about a well-defined value, which is greatly enhanced compared to the noninteracting prequench state. For quenches to large attractive interaction strengths $|\gamma| \gtrsim 10$, single-frequency oscillations in the local second-order correlation function on top of an overall irregular behaviour are observed, with the oscillations persisting at late times. The oscillatory behaviour also occurs in the momentum distribution for large postquench interaction strengths, and the frequency of oscillation is determined by the energy difference between the dominant super-Tonks eigenstate and

the most highly occupied two-body bound state following the quench. Similar oscillations in the local third-order correlation function occur at a frequency given by the energy difference between two- and three-body bound states of the postquench Hamiltonian.

Time-averaged values of the postquench local second-order correlation function appear consistent with a tendency towards a constant value in the limit of infinitely strong attractive interactions. In particular, our results for this quantity indicate an enhancement by a factor of $\simeq 4$ over the initial ideal-gas value, consistent with a recently obtained thermodynamic-limit result [35, 36]. Our calculations similarly suggest that the time-averaged local third-order correlation function following the quench tends to a constant, greatly enhanced value in the strongly interacting limit. Outside interparticle separations of the order of the extent of bound states of the Lieb–Liniger model, the dynamical behaviour and time-averaged form of the second-order correlation function following a quench to attractive interactions are remarkably similar to those following a quench to repulsive interactions of the same magnitude.

## Acknowledgements

M.J.D. acknowledges the support of the JILA Visiting Fellows program.

**Funding information** This work was partially supported by ARC Discovery Projects, Grant Nos. DP110101047 (J.C.Z., T. M.W., K.V. K., and M.J.D.), DP140101763 (K.V. K.), DP160103311 (M.J.D.) and by the EU-FET Proactive grant AQuS, Project No. 640800 (T.G.).

## A Mean-field correlation functions

In this appendix we describe how we obtained the mean-field results for comparison with the Lieb–Liniger results plotted in Figs. 1 and 2. The solution of the 1D Gross–Pitaevskii equation on a ring of finite circumference $L$ is conveniently expressed in terms of the angular coordinate $\theta \in [0, 2\pi)$ around the ring circumference (see e.g. Refs. [44, 75]) as

$$\Psi_{\text{GP}}(\theta, \Theta) = \begin{cases} \sqrt{\frac{1}{2\pi}}, & \gamma^{(r)} \geq \gamma^{(r)}_{\text{crit}}, \\ \sqrt{\frac{K(m)}{2\pi E(m)}} \text{dn}\left(\frac{K(m)}{\pi}(\theta - \Theta)\Big| m\right), & \gamma^{(r)} < \gamma^{(r)}_{\text{crit}}, \end{cases} \tag{15}$$

where $\gamma^{(r)} = \gamma N^2/(2\pi^2)$ is the interaction strength, $\Theta$ is the centre of the soliton, and we have assumed periodic boundary conditions $\Psi_{\text{GP}}(0) = \Psi_{\text{GP}}(2\pi)$. In these units the critical value of the interaction strength is $\gamma^{(r)}_{\text{crit}} = -0.5$. The functions $K(m)$ and $E(m)$ are the complete elliptic integrals of the first and second kind, respectively, and $\text{dn}(x|m)$ is one of the Jacobian elliptic functions. The parameter $m \in [0, 1]$ is fixed by the solution to

$$K(m)E(m) = \frac{\pi^2 \gamma^{(r)}}{2}. \tag{16}$$

The Gross–Pitaevskii equation arises by approximating the many-body wave function using a Hartree-Fock ansatz $\Psi(\theta_1, \ldots, \theta_N) = \prod_{j=1}^{N} \Psi_{\text{GP}}(\theta_j, \Theta)$, where the single-particle wave function depends on the centre-of-mass variable $\Theta$ (15). Following Ref. [80], we restore the translational symmetry of the many-body wave function by taking a coherent superposition of symmetry-broken Gross–Pitaevskii states with different soliton locations

$$\Psi(\theta_1, \ldots, \theta_N) = \frac{1}{\sqrt{2\pi}} \int_0^{2\pi} d\Theta^N \prod_{j=1}^{N} \Psi_{\text{GP}}(\theta_j, \Theta). \tag{17}$$

The normalized correlation functions are then given by

$$g^{(1)}(\theta, \theta') = \frac{G^{(1)}(\theta, \theta')}{\sqrt{G^{(1)}(\theta, \theta)G^{(1)}(\theta', \theta')}},$$

$$g^{(2)}(\theta, \theta') = \frac{G^{(2)}(\theta, \theta')}{G^{(1)}(\theta, \theta)G^{(1)}(\theta', \theta')}, \tag{18}$$

where

$$G^{(1)}(\theta, \theta') = \frac{N}{2\pi} \int_0^{2\pi} d\Theta \, \Psi_{\text{GP}}^*(\theta, \Theta)\Psi_{\text{GP}}(\theta', \Theta), \tag{19}$$

and similarly

$$G^{(2)}(\theta, \theta') = \frac{N(N-1)}{2\pi} \int_0^{2\pi} d\Theta \, \Psi_{\text{GP}}^*(\theta, \Theta)\Psi_{\text{GP}}(\theta, \Theta)\Psi_{\text{GP}}^*(\theta', \Theta)\Psi_{\text{GP}}(\theta', \Theta). \tag{20}$$

## B  Details of numerical algorithm for finding eigenstates with bound states

Eigenstates with complex rapidities arrange themselves in so-called string patterns in the complex plane for large values of $|c|L \equiv N|\gamma|$, up to deviations from these strings that are exponentially small in the system size $L$ at fixed $|c|$ [23, 39, 42, 43, 74]. This requires a reformulation of the algorithm previously described in Ref. [33] so as to avoid a loss of numerical accuracy due to calculating the difference between two nearly equal values. In this appendix we describe the the details of this procedure for $N = 2, 3$, and 4 particles. Extending this procedure to $N > 4$ particles is possible, but the number of factors that have to be considered increases rapidly with increasing $N$.

### B.1  $N = 2$ particles

We begin by considering the $N = 2$ particle ground state, for which the rapidities are imaginary for all $c < 0$. For intermediate and large $|c|L$ the rapidities in this case are

$$\lambda_j = \mp i\frac{c}{2} + i\delta_j, \tag{21}$$

where the minus (plus) sign applies to $\lambda_1$ ($\lambda_2$) by convention. The string deviations $\delta_j \propto e^{-\eta L}$, where $\eta$ is a positive constant. The (unnormalized) two-particle wave function reads

$$\zeta(x_1, x_2) = (\lambda_2 - \lambda_1 - ic)e^{i(\lambda_1 x_1 + \lambda_2 x_2)} - (\lambda_1 - \lambda_2 - ic)e^{i(\lambda_2 x_1 + \lambda_1 x_2)},$$

$$\equiv -i\left[(2\lambda + c)e^{\lambda r} + (2\lambda - c)e^{-\lambda r}\right], \tag{22}$$

where we defined the relative coordinate $r = x_2 - x_1$ and $\lambda = \lambda_1/i = -\lambda_2/i$. In light of Eq. (21), the first term in the last line of Eq. (22) is a product of a small number $(2\lambda + c)$ and a large number $(e^{\lambda r})$ away from $r = 0$. The former is a difference of two numbers that are nearly equal, leading to catastrophic cancellations in double-precision arithmetic. However, from Eqs. (6) and (21) we find

$$2\lambda + c \equiv 2\delta_1 = e^{-\lambda L}(2\lambda - c), \tag{23}$$

and substituting this expression into Eq. (22) renders it amenable to numerical evaluation.

## B.2 $N = 3$ **particles**

For particle numbers $N > 2$, in addition to the ground state, which always has imaginary rapidities, excited parity invariant states may possess complex rapidities at interaction strengths $c < c_{\mathrm{crit}}$, where $c_{\mathrm{crit}}$ is an $N$-dependent "phase-crossover" point in the vicinity of the mean-field transition point [39]. For $N = 3$, there are two parity-invariant eigenstates with complex rapidities:

(i) The ground state is a three-body bound state with imaginary rapidities $\lambda_1 = -\lambda_3$, and $\lambda_2 = 0$. By convention $\lambda_1/i > 0$. For small string deviations, the factor $\lambda_2 - \lambda_1 - ic \equiv -(\lambda_1 + ic)$ needs to be rewritten. The Bethe equation (6) for $\lambda_1$ is

$$e^{i\lambda_1 L} = \frac{\lambda_1 + ic}{\lambda_1 - ic} \frac{2\lambda_1 + ic}{2\lambda_1 - ic}, \tag{24}$$

which can be rearranged to find an expression

$$\lambda_1 + ic = e^{i\lambda_1 L}(\lambda_1 - ic)\frac{2\lambda_1 - ic}{2\lambda_1 + ic} \tag{25}$$

for the critical factor in this case.

(ii) First excited parity invariant state. Here, the rapidities $\lambda_1 = -\lambda_3$ are real for $c > c_{\mathrm{crit}}$ [39] and are otherwise imaginary, in which case we again follow the convention that $\lambda_1/i > 0$. The critical factor to be replaced is $2\lambda_1 + ic$. From Eq. (25) we obtain the appropriate expression

$$2\lambda_1 + ic = e^{i\lambda_1 L}(2\lambda_1 - ic)\frac{\lambda_1 - ic}{\lambda_1 + ic}. \tag{26}$$

## B.3 $N = 4$ **particles**

For $N = 4$ particles, an infinite number of parity-invariant bound states contribute to the postquench dynamics, and they can be grouped into four different categories, cf. Sec. 4.1. In the following we write $\lambda_j \equiv \mu_j + i\nu_j$ with $\mu_j, \nu_j$ real numbers, and assume that $\mu_1, \mu_2 \geq 0$, $\nu_1, \nu_2 \geq 0$, $\lambda_3 = -\lambda_2$, and $\lambda_4 = -\lambda_1$.

(i) The ground state with $\{n_j\} = \{0,0\}$. The rapidities are purely imaginary, $\mu_j = 0$. Substituting this into Eq. (6) leads to the following two equations.

$$e^{-\nu_1 L} = \frac{\nu_1 - \nu_2 + c}{\nu_1 - \nu_2 - c} \frac{\nu_1 + \nu_2 + c}{\nu_1 + \nu_2 - c} \frac{2\nu_1 + c}{2\nu_1 - c}, \tag{27}$$

$$e^{-\nu_2 L} = \frac{\nu_2 - \nu_1 + c}{\nu_2 - \nu_1 - c} \frac{\nu_2 + \nu_1 + c}{\nu_2 + \nu_1 - c} \frac{2\nu_2 + c}{2\nu_2 - c}. \tag{28}$$

There are two critical factors: $\nu_1 - \nu_2 + c$ and $2\nu_2 + c$. Rewriting Eq. (27) leads to

$$\nu_1 - \nu_2 + c = e^{-\nu_1 L}(\nu_1 - \nu_2 - c)\frac{\nu_1 + \nu_2 - c}{\nu_1 + \nu_2 + c} \frac{2\nu_1 - c}{2\nu_1 + c} \equiv \alpha. \tag{29}$$

Equation (28) can be expressed as

$$2\nu_2 + c = -e^{-\nu_2 L}\alpha \frac{\nu_2 + \nu_1 - c}{\nu_2 + \nu_1 + c} \frac{2\nu_2 - c}{\nu_2 - \nu_1 + c}, \tag{30}$$

where $\alpha$ is the first critical factor defined in Eq. (29).

(ii) The three-body bound state with $\{n_j\} = \{1,0\}$. This is the first parity invariant excited state and has real rapidities $\lambda_1$ and $\lambda_4$ that tend to zero for large attractive values of $cL$. Following Ref. [28], Appendix B, we can reparameterize the rapidities in this case via their deviations $\delta = e^{-|c|L/2}$ from the string solution

$$
\begin{aligned}
\lambda_1 &= \delta\alpha\,, \\
\lambda_2 &= -ic + i\delta^2\beta\,.
\end{aligned}
\tag{31}
$$

Substituting this into the Bethe equations (6), Ref. [28] obtained in the limit of small string deviations

$$
\begin{aligned}
\alpha &= \sqrt{12}\,|c|\,, \\
\beta &= 6Lc^2\,.
\end{aligned}
\tag{32}
$$

We did not find a suitable double-precision strategy for this particular eigenstate, and so resorted to high-precision arithmetic for numerical calculations. To obtain sufficiently precise Bethe rapidities for large attractive values of $\gamma$, we used Eqs. (32) as the starting point for our root-finding algorithm.

(iii) Eigenstates with $\{n_j\} = \{n,0\}$ for all integers $n \geq 2$. In this case, $\lambda_1$ is real, $\lambda_2$ imaginary, $\lambda_1 = \mu_1$, $\lambda_2 = i\nu_2$. The critical factor is $2\nu_2 + c$. Rewriting the Bethe equation for $\lambda_2$ leads to

$$
2\nu_2 + c = e^{-\nu_2 L}(2\nu_2 - c)\frac{|\mu_1 + i(\nu_2 - c)|^2}{|\mu_1 + i(\nu_2 + c)|^2}\,.
\tag{33}
$$

(iv) Eigenstates with $\{n_j\} = \{n,n\}$ for all integers $n \geq 1$. The Bethe rapidities are complex and satisfy $\lambda_1 = \lambda_2^*$. Rewriting the first Bethe equation with $\mu \equiv \mu_1 = \mu_2$ and $\nu \equiv \nu_1 = -\nu_2$ and taking the real part leads to

$$
2\nu + c = \frac{2\nu - c}{2\mu}e^{-\nu L}\,\Re\left[(2\mu + i(2\nu - c))\frac{2\mu - ic}{2\mu + ic}e^{i\mu L}\right],
\tag{34}
$$

where $\Re[x]$ denotes the real part of $x$.

(v) Eigenstates with $\{n_j\} = \{n,n-1\}$ for all integers $n \geq 2$. For $c > c_{\text{crit}}$, the Bethe rapidities are real. For more attractive interactions, they become complex conjugate pairs, $\lambda_1 = \lambda_2^*$, and this case becomes equivalent to the preceding one.

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
