# Peer review of "Quantum quench dynamics of the attractive one-dimensional Bose gas via the coordinate Bethe ansatz"

_SciPost Physics, doi:SciPost Phys. 4, 011 (2018)_

## Round 1 · Referee Report · Anonymous (Referee 1) · 2017-6-21

Strengths

  1. Interesting and timely results
  2. Detailed analysis with well explained physics

Weaknesses

No major ones

Report

The authors study correlation functions in the attractive regime of the integrable Lieb-Liniger model. They compute the ground state and the post-quench correlation functions for systems with few particles using the coordinate Bethe ansatz wavefunction and the methods the authors developed in [32] and [33]. In this work they extend the applicability of their method from repulsive to attractive interactions. The main point of the paper is to show how the presence of the bound states in the spectrum of the attractive Lieb-Liniger gas influences the correlation functions and, as a consequence, how they differ from those of the repulsive gas.

The results are interesting and authors put a lot of attention to highlight the physics behind them. When possible the results are compared with previous works.

The results are presented in a clear way and the paper is well written. I have only few minor suggestions for improvements.

Requested changes

  1. I think the presentation would benefit from dividing subfigures of figures 1,2 and 11 into separate figures and placing them closer to where they are referred to in the text. It's just a suggestion.

  2. On page 9 the authors write: “In Fig. 1(a) we compare our exact results to the mean-field solution just on the localized side of the crossover at γ = −0.21 (green crosses), and find that the exact many-body solution (green dotted line) is slightly more localized.” Is there an explanation why the exact solution is more localized. Naively one could expect the quantum fluctuations to delocalize the exact solution as compared with the mean-field result.

  3. On page 10 the authors write “We note that the momentum distributions for the most strongly interacting systems considered here are much broader than those of ground states in the strongly repulsive regime (cf., e.g., Ref. [33]).” Could authors provide a bit more details here instead of just referring to the previous work?

  4. On page 24 the authors write “For comparison, we also plot the constant ideal-gas form $g^{(2)}(x, t = 0) = 1 − 1/N$ of this function in the initial state”. Wouldn’t it be worth to compare the post-quench results with the $g^{(2)}$ of the ground state with final value of the interactions instead?

  • validity: high
  • significance: good
  • originality: good
  • clarity: high
  • formatting: excellent
  • grammar: perfect

Author:  Matthew Davis  on 2017-11-17  [id 191]

(in reply to Report 1 on 2017-06-21)
Category:
answer to question

We appreciate the time that the referee has invested in reading our manuscript and writing the report. Our responses to the requested changes are below.

Requested changes: 1. I think the presentation would benefit from dividing subfigures of figures 1,2 and 11 into separate figures and placing them closer to where they are referred to in the text. It's just a suggestion.

Response: While we appreciate the suggestion, all the subplots in each of these figures are closely related to one another, and therefore we believe they are best presented together. Currently they are all within a page of where they are discussed in the text - breaking them up wouldn’t help this a lot. Additionally, there are already quite a few figures in the manuscript, and we believe that creating 8 more would potentially make the manuscript more difficult to read.

  1. On page 9 the authors write: “In Fig. 1(a) we compare our exact results to the mean-field solution just on the localized side of the crossover at $\gamma = −0.21$ (green crosses), and find that the exact many-body solution (green dotted line) is slightly more localized.” Is there an explanation why the exact solution is more localized. Naively one could expect the quantum fluctuations to delocalize the exact solution as compared with the mean-field result.

Response: The referee is correct that we would expect the mean-field solution to be more localised in general. The opposite behaviour observed here for $\gamma=-0.21$ appears to be due to finite-size broadening of the crossover, as can be seen from close inspection of the entropies plotted in the inset to Fig. 1(d). We have added the following text on this point: "We note that this behaviour is consistent with that of the entanglement entropy [inset to Fig. 1(d)], which is smaller for the exact solution than for the mean-field approximation for $|\gamma| \gtrsim 0.23$. By contrast, at weaker interaction strengths finite-size rounding of the crossover yields an entropy for the exact system larger than the mean-field value."

  1. On page 10 the authors write “We note that the momentum distributions for the most strongly interacting systems considered here are much broader than those of ground states in the strongly repulsive regime (cf., e.g., Ref. [33]).” Could authors provide a bit more details here instead of just referring to the previous work?

Response: Thank you for pointing this out. We have amended the text to state: "We note that the momentum distributions for the most strongly interacting systems considered here are much broader than the "hump" that forms in the ground-state momentum distribution of the repulsive gas in the strongly interacting Tonks limit, which extends to $\simeq 2k_F$ [33, 87, 88]."

  1. On page 24 the authors write “For comparison, we also plot the constant ideal-gas form g(2)(x,t=0)=1−1/N of this function in the initial state”. Wouldn’t it be worth to compare the post-quench results with the g(2) of the ground state with final value of the interactions instead?

Response: The purpose of plotting this ideal-gas value is largely so the reader can see where g2(x) has been enhanced or suppressed relative to the initial state. As the initial value is 1-1/N=0.75 rather than simply 1, we think it is worthwhile to have it indicated here. Comparison to the ground state at the final $\gamma$ would perhaps not be so useful -- recall from Fig. 2 that at $\gamma=-40$ we have g2(0)=100! It might be more enlightening to compare to thermal values of the correlation function corresponding to the conserved energy, as we have done in Refs. [32,33]. However, calculating these quantities was computationally prohibitive in the attractive case. We have amended the text to read: "In Fig. 11(b), we plot the second-order correlation function gDE(2)(x) in the diagonal ensemble for several postquench interaction strengths $\gamma$ and compare these to the initial-state form g(2)(x,t=0)=1−1/N of this function (horizontal line)."

---

## Round 1 · Referee Report · Anonymous (Referee 2) · 2017-6-26

Strengths

1- The paper uses an exact method to compute the correlation functions of a mesoscopic sample of few bosons following a quantum quench to attractive interactions 2- The topic is very timely and the subject will be of interest for experimentalist in cold atoms 3- This work complements previous results in the domain 4- The paper is clearly written, and the references are complete

Weaknesses

1- The paper is very extensively described : all the cases and sub-cases are considered, and the figures explained in detail, but it is not easy to grasp what is the main physical conclusion.

Report

The paper provides a detailed analysis of the behaviour of a system of 1D bosons with attractive interactions. In particular, after obtaining the equilibrium correlation functions, the behaviour following a quench to negative interactions is studied. The paper is a step forward in the understanding strongly correlated one-dimensional quantum systems, and complements previous studies in the same direction. The paper provides a very complete analysis. I would suggest to put more emphasis in the text on the main results, as well as more physical comments wherever possible. In my opinion, the result of Fig6 is very nice and would deserve to be well put into visibility.

Requested changes

1- The authors should make a communication effort, putting in value in the text the most important results (and maybe put some benchmarking results in appendices?). In this way the paper would become accessible to a broader audience, and directly target experimentalists.

2- In the introduction, I suggest to make a more explicit connection between the current work and the previous work by the same authors Ref 33, as well as to explain more in detail their contribution wrt to previous works in Refs [35-36]

3- Pag 9, the authors mention that the lowest nonzero momentum modes start to deviate from the $k^{-4}$ scaling. Why is this surprising? The scaling holds only at large k. The authors should explain what they meant.

3-Pag 11, the authors state that at stronger interactions the agreement with mean-field theory are in closer agreement with their results. Can the authors state why? Is it coincidental? One would expect that mean-field theory to hold only at weak interactions.

4- Pag 15 The authors mention a monochromatic oscillatory behaviour for a quench to $\gamma=-0.5$. This is not easy to check since the figure is in log time scale. Can the authors provide an inset at short times in linear scale?

5- Pag 16 The authors notice that $g^3$ is less noisy than $g^2$. Is there any explanation for this?

6- Fig 7: the long-times behaviour look very noisy. Is it an artefact of the log scale? Or what is the reason for that? Should we trust the curves at such times?

  • validity: high
  • significance: good
  • originality: high
  • clarity: high
  • formatting: good
  • grammar: excellent

Author:  Matthew Davis  on 2017-11-17  [id 192]

(in reply to Report 2 on 2017-06-26)

We appreciate the time that the referee has invested in reading our manuscript and writing the report, and the positive comments Our responses to the requested changes are below.

Requested changes: 1- The authors should make a communication effort, putting in value in the text the most important results (and maybe put some benchmarking results in appendices?). In this way the paper would become accessible to a broader audience, and directly target experimentalists.

Response: We respond to this first point in the requested changes with reference to the issue raised by the referee in "Weaknesses": that it is not easy to grasp the main physical conclusion of the research. We would like to point out that our purpose in this paper is, as is often the case in theoretical/computation papers of this nature, primarily to report the results of our calculations. We submit, respectfully, that it is not essential that such papers have a "main physical conclusion". We believe the contribution of the paper should be clear enough: we have applied an essentially exact methodology to investigate the nonequilibrium correlation functions resulting from this particular quench scenario, in the limiting case of small particle numbers to which our approach is applicable. We have, where possible, contrasted the behaviour of this system to the more well-established behaviour of the Lieb-Linger gas following a quench to repulsive interactions, and compared our results to those of other approaches to this problem where such exist -- namely, the results for the post-quench values of g2 found by Piroli et al. (Refs. [35,36]).

We note that all three referees have indicated the clarity with which our results are reported, and referees one and two have both remarked positively on the level of detail with which our results are described. While we may be missing an opportunity to promote our work more vigorously to, e.g., experimentalists in this area, we believe that the quality of communication of results in the paper is of a standard sufficient for publication, consistent with many of the remarks made by the three referees.

Having said that, we do agree that it would be worthwhile to include more text to summarise and highlight the key results of Sections 4 and 5 at the ends of these sections. We have expanded on the summary at the end of Section 4 to write:

"In summary, quenches from the ideal-gas ground state to attractive values of \gamma result in the occupation of energy eigenstates containing bound states in addition to the gas-like scattering states of the attractively interacting model, which are analogous to the eigenstates of the repulsively interacting Lieb–Liniger gas. As the magnitude |\gamma| of the final interaction strength is increased, the postquench occupations of the gas-like excited states approach those of their counterparts following a quench to the corresponding repulsive interaction strength, and the occupations of bound states eventually decrease. However, these bound states significantly influence the dynamics of postquench correlation functions for all final interaction strengths we have considered, causing large oscillations in local correlations and in the occupation of the zero-momentum mode. For large attractive values of \gamma, bound states are highly localized and thus influence the second-order correlation function only at small separations, whereas at larger separations this function exhibits postquench dynamics similar to those observed following quenches to repulsive interactions [32].

We have added also the following summary paragraph at the end of Section 5:

"In summary, our results for the time-averaged local second-order correlation function g(2)DE(0) are consistent with an enhancement of this quantity over the initial ideal-gas value by a factor of \simeq 4 in the limit of strong final interaction strengths, and thus with the predictions of Refs. [35, 36] in this limit. Our calculations also reveal an enhancement of the local third-order correlation function g(3)DE(0) over the ideal-gas value by a factor of \simeq 20 for strong interactions, suggesting that the postquench state would be susceptible to large three-body recombination losses in practice. Results for time-averaged correlation functions at interparticle separations larger than the characteristic extent of bound states are comparable to those obtained previously [32] for quenches to repulsive interactions."

We believe that these revisions to the text, together with the additions made in response to the referee's second point, described below, make the central results of our work clearer.

2- In the introduction, I suggest to make a more explicit connection between the current work and the previous work by the same authors Ref 33, as well as to explain more in detail their contribution wrt to previous works in Refs [35-36].

Response: Thank you for this suggestion. We have revised and extended the introduction. It now reads in part:

"In Refs. [32, 33] we developed a methodology for the calculation of equilibrium and nonequilibrium correlation functions of the repulsively interacting Lieb-Liniger gas based on the semi-analytical evaluation of matrix elements between the eigenstates of the Lieb-Liniger Hamiltonian given by the coordinate Bethe ansatz. Here we extend this approach to the attractively interacting gas, for which the Bethe rapidities that characterize the eigenstates are in general complex-valued, indicating the presence of multiparticle bound states. We apply our method to calculate results for the time evolution of correlation functions following a quench to attractive interactions from the ideal-gas ground state for a system of four particles. As in our previous studies of quenches to repulsive interactions [32, 33], we find that finite-size effects are significant for quenches to weak final interaction strengths. For strong final interaction strengths our results for the time-averaged local second-order correlation function are consistent with the stationary values in the thermodynamic limit calculated in Refs. [71, 72]. In contrast to that work, however, our approach allows us to also calculate the time-averaged value of the postquench third-order correlation function, which we find to be dramatically enhanced over the ideal-gas value, implying that three-body recombination losses would be significant in experimental realizations of the quench. Our approach also allows us to calculate the dynamical evolution of correlation functions following the quench, and for a quench to strong attractive interactions we observe behaviour similar to that following a quench to repulsive interactions of the same magnitude, superposed with characteristic contributions of bound states at small interparticle separations."

We hope the referee agrees that this both makes more explicit the connection between the current work and our previous Refs. [32,33], and explains the contribution of this work with respect to the results of Piroli et al. [35,36].

3- Page 9, the authors mention that the lowest nonzero momentum modes start to deviate from the $k^{-4}$ scaling. Why is this surprising? The scaling holds only at large k. The authors should explain what they meant.

Response: We simply meant to point out that for the weakest interaction strengths we consider, this scaling effectively persists for all nonzero k in the finite-size geometry we consider, and that as interactions are increased less trivial structure of n(k) is observed. We have revised the text to state: "For the case of $\gamma = −0.1$ (red empty circles), interactions are sufficiently weak that no visible deviation from this scaling is visible at the smallest nonzero momenta $k_j$ resolvable in our finite geometry. By contrast, for $\gamma = −0.21$ (green triangles), less trivial behaviour of the momentum distribution can be seen, with the lowest nonzero momentum modes deviating visibly from the $\propto k^{-4}$ scaling."

4- Page 11, the authors state that at stronger interactions the agreement with mean-field theory are in closer agreement with their results. Can the authors state why? Is it coincidental? One would expect that mean-field theory to hold only at weak interactions.

Response: We do not have a precise argument for this point, so we have not added anything to the manuscript text. We do not think it is simply coincidental, however. In the case of a repulsive Bose gas mean-field theory degrades as the interaction strength is increased, due to particles being scattered out of the mean-field wavefunction (i.e., a uniform condensate). For the correlated cluster-like states of the attractive Bose gas we consider here, this picture no longer applies. The mean-field wavefunction in this case corresponds to all particles residing in the same single-particle state -- as in the usual Gross-Pitaevskii/Hartree approximation -- except that this state is now localized in space (in a symmetry-broken picture). As the interaction strength is increased the particles in the exact N-particle wavefunction become more tightly bound together, and this wavefunction apparently comes to more closely resemble the Hartree product of localized states, as the attractive interactions cause the particles to effectively "condense" together in real space. (In the finite system the wavefunction is of course in both cases a symmetry-restored superposition of these localized states, but apparently the exact symmetric wavefunction also agrees more closely with the symmetrized mean-field wavefunction as the interaction strength is increased.)

5- Page 15 The authors mention a monochromatic oscillatory behaviour for a quench to $\gamma=-0.5$. This is not easy to check since the figure is in log time scale. Can the authors provide an inset at short times in linear scale?

Response: With respect, we do not think it is worthwhile to add an inset showing the short-time behaviour here, given that the figure panels are already quite dense and that this is a relatively unimportant point. We hope the referee would agree that there would be little to be gained by attempting to mislead the reader on a point of such little consequence (indeed, all it really serves to show is that the dynamics of our finite-size system are uninteresting for such weak interactions). Moreover, we were careful to say that the behaviour is "nearly" monochromatic, and that this is the case should be sufficiently clear from the log-scale plot as it is. We have, however, revised the text later in the paragraph from "determines the frequency" to "determines the dominant frequency" for consistency.

6- Page 16 The authors notice that $g^3$ is less noisy than $g^2$. Is there any explanation for this?

Response: We have rewritten the text to say: "For larger attractive values of the postquench interaction strength, on the other hand, the shape of g(3)(0,t) is more regular compared to g(2)(x=0,t), reflecting the fact that only one three-body bound state contributes to the postquench wavefunction, whereas multiple states containing bound pairs are present."

7- Fig 7: the long-times behaviour look very noisy. Is it an artefact of the log scale? Or what is the reason for that? Should we trust the curves at such times?

Response: We believe that this appearance results, as the referee suggests, simply from the log scale, and probably the particular linestyles used. It is perhaps not pretty but we believe it communicates the important points clearly enough. We do expect that the curves can be trusted at these late times, as only weakly occupied (and high energy) eigenstates are neglected in calculating these curves. The effects of these omitted contributions would therefore be very small, high frequency perturbations to the plotted curves, and it is unlikely these corrections would even be visible on the scales of these plots.

---

## Round 1 · Referee Report · Anonymous (Referee 3) · 2017-7-6

Strengths

1- Considers a rich phase of the Lieb-Liniger model. 2- Estimates the particle losses in the quench from the ideal-gas ground state to the attractive Lieb-Liniger.

Weaknesses

1- The number of particles considered in out-of-equilibrium situations is very low.

Report

The paper uses coordinate Bethe ansatz to study the attractive phase of the Lieb-Liniger model in finite volume with a finite number of particles. The authors study three specific correlation functions and the momentum distribution, both in the ground state of the attractive chain and after a quench from the ideal-gas ground state to the attractive regime.

Studying the ground state, the authors find that increasing the attractive interaction strength $|\gamma|$ the correlations become more and more peaked. Eventually, they get to be well described by the "bright-soliton" mean field solution; also the crossover interaction strength appears compatible with the mean-field result.

In their non-equilibrium study the authors compare the time evolution of correlations after two distinct quenches. (1) From the ideal-gas ground state to the attractive regime. (2) From the ideal-gas ground state to the repulsive regime. These quenches are chosen such that the absolute value of the dimensionless interaction $\gamma$ is the same. They find that the effect of bound states, present only for attractive interactions, is stronger for $\gamma<0$ small. For large attractive interactions $\gamma$ the dynamics is very close to that generated by $|\gamma|$. The only important difference is found by looking at expectation values of operators at the same point, e.g., $\langle{\hat\Psi^\dagger(0,t)^n\hat\Psi(0,t)^n}\rangle$. These expectation values are much larger for attractive interactions. In particular, this is true for $g^{(3)}(0,t)=\langle{\hat\Psi^\dagger(0,t)^3\hat\Psi(0,t)^3}\rangle$, which can be used as a measure for particle losses in the experiments.

I think that the paper is interesting. It considers a phase of the model characterised by rich physics and gives a thorough analysis based on the numerical solution of the exact Bethe equations. Therefore, I recommend the publication of this paper on Scipost. The main weakness of the paper lies in the small number $N$ of particles that the authors can consider. Both in the ground state studies for large attractive interactions and in the out-of-equilibrium setting they consider at most $N=4$. Considering larger numbers of particles would be, e.g., very useful to give more quantitative estimations of the particle losses through $g^{(3)}(0,t)$ and investigate the experimental realisability of the quench considered.

Requested changes

Minor questions/comments

  • I find the notation for the eigenstates of the Hamiltonian slightly confusing, in particular in Section 4.1. Maybe it would be better to use a notation which makes more transparent the presence of bound rapidities in the state.

  • Why do the authors not consider the time evolution of the two point function $g^{(1)}(x,t)$ after the quench?

  • validity: high
  • significance: good
  • originality: good
  • clarity: high
  • formatting: excellent
  • grammar: excellent

Author:  Matthew Davis  on 2017-11-17  [id 193]

(in reply to Report 3 on 2017-07-06)

We appreciate the time that the referee has invested in reading our manuscript and writing the report. We would liked to have performed the calculations for larger numbers of particles, but even going to $N=5$ is prohibitive, due to the numerical difficulties as described in appendix B. Our further responses to the requested changes are below.

Requested changes:

Minor questions/comments

  • I find the notation for the eigenstates of the Hamiltonian slightly confusing, in particular in Section 4.1. Maybe it would be better to use a notation which makes more transparent the presence of bound rapidities in the state.

Response: We appreciate the referee's point, but it would not be so straightforward to introduce a new notation to address this, particularly as we are using the same quantum number convention to label states of the attractive and repulsive system. We would have to somehow retain the current quantum numbers and introduce an additional notation indicating, e.g., bound states also, and we think that on balance doubling up the notation like this would probably make the presentation less, rather than more, clear. Where the structure of the state (i.e., as a two- or three-body bound state, or a gas-like state) is important we do indicate this in the text alongside the quantum number pair labelling the state. It is perhaps not as elegant as it could be, but we believe all the important information is communicated unambiguously.

  • Why do the authors not consider the time evolution of the two point function g(1)(x,t) after the quench?

Response: We do not show the time evolution of g(1)(x,t) mainly because of the computational expense of calculating this quantity. As compared to g2(x,t), the first-order correlation function requires an additional integration over space, and thus an additional combinatorial increase in the number of terms involved. It would therefore be very expensive to obtain a high-resolution coordinate-space representation of g(1)(x,t) analogous to the one we present for g(2)(x,t). The most important features of the first-order correlations are shown in our plots of the momentum distribution n(k,t), which is perhaps a more useful physical picture anyway. The calculation of this quantity is more tractable as we only plot the first six momentum modes, which can be evaluated exactly from a comparatively sparse spatial representation of g1(x,t).

---

## Round 2 · Referee Report · Anonymous (Referee 1) · 2017-12-6

Report

I'm satisfied with the authors' response and changes in the manuscript.

---

## Round 2 · Referee Report · Anonymous (Referee 3) · 2017-12-12

Report

I am satisfied with the current version of the paper, I think that the authors made all the possible efforts to improve it and the numerical results presented are the best they can produce. So I recommend this paper for publication in Scipost.

---

## Round 2 · Author Response

Dear editor,

We appreciate the time and effort the three referees have invested in reading our manuscript, and the resulting suggestions for clarification and improvement. We have made changes in response to the majority of these suggestions, and explained our reasons for not making changes for those that remain in our responses to the reports. We believe that the manuscript has improved as a result.

We apologise for the length of time it has taken us to make the revisions. We hope that our work can now be accepted for publication.

Best regards,
Jan Zill, Tod Wright, Karen Kheruntsyan, Thomas Gasenzer, Matthew Davis.

---

## Round 2 · List of Changes

Warnings issued while processing user-supplied markup:

  • Inconsistency: Markdown and reStructuredText syntaxes are mixed. Markdown will be used.
    Add "#coerce:reST" or "#coerce:plain" as the first line of your text to force reStructuredText or no markup.
    You may also contact the helpdesk if the formatting is incorrect and you are unable to edit your text.

Summary of changes in response to referee reports

(full details in our individual responses to the reports)

  • Expanded the introduction to make a more explicit connection with our earlier work, and more detail about the contribution of this paper relative to others.

  • Added further explanation regarding the hump in momentum distributions for strongly interacting systems in Fig 2(c).

  • Added a summary to the end of section 4 and section 5 to highlight the key results.

  • Added text explaining the reason for the mean-field solution for \gamma = −0.21 in Fig 1(a) being broader than exact solution.

  • Revised the text regarding the k^-4 scaling of the momentum in Fig. 2(c).

  • Commented on the more regular nature of $g^3$ relative to $g^2$ in Fig 5.

  • Expanded on the comparison with the ideal gas correlation function in Fig. 11(b).

Additional revisions to the manuscript:

  • Sec. 1: We removed a citation to "G. Dvali et al., Scrambling in the black hole portrait" as upon revisiting this we decided it does not bear on the point we are making in the text.

  • Sec. 3.1: We corrected references to "pink diamonds" in Fig. 1 to "pink triangles".

  • Sec. 4.2, we replaced: "energy difference between the two-body bound state {n_j} = {2,0} and the three-body bound state {n_j} = {1,0}" with "energy difference between the three-body bound state {n_j} = {1,0} and the predominant two-body bound state {n_j} = {2,0}".

  • Sec. 5, below Eq. (14), we added a sentence: "We note that in practice the sum in Eq. (14) runs over a finite set of energy eigenstates with populations |C_{lambda_j}|^2 exceeding some threshold value." (As although this detail should be clear enough at this point, it should be noted explicitly as in our previous Refs. [32, 33].)

  • Fig. 10, we replaced "strong-coupling thermodynamic-limit prediction for g(2)DE(0)" with the more accurate "strong-coupling (order-1/gamma^3) thermodynamic-limit prediction for the stationary value of g(2)(0)".

  • Appendix A: We have slightly reordered the text around equations (15)-(17) for clarity and everywhere replaced the symbol theta_0 with a capital Theta.

  • Appendix B: We added citations to references [42,43] for the concept of the string deviations.

  • Appendix B.3, above Eq. (32) we added the words "in the limit of small string deviations" to make this point more clear (and accurate).

  • We also made a few other very minor changes to wording and punctuation throughout the manuscript to make the presentation more clear.

---

## Editorial Decision

published